# Learning Interpretable Low-dimensional Representation via Physical Symmetry

**Xuanjie Liu**[1,2]        **Daniel Chin**[1,2]        **Yichen Huang**[1]        **Gus Xia**[1,2]

[1]Mohamed bin Zayed University of Artificial Intelligence
[2]New York University Shanghai
{Xuanjie.Liu, Nanfeng.Qin, Yichen.Huang, Gus.Xia}@mbzuai.ac.ae

## Abstract

We have recently seen great progress in learning interpretable music representations, ranging from basic factors, such as pitch and timbre, to high-level concepts, such as chord and texture. However, most methods rely heavily on music domain knowledge. It remains an open question what general computational principles *give rise to* interpretable representations, especially low-dim factors that agree with human perception. In this study, we take inspiration from modern physics and use *physical symmetry* as a self-consistency constraint for the latent space of time-series data. Specifically, it requires the prior model that characterises the dynamics of the latent states to be *equivariant* with respect to certain group transformations. We show that physical symmetry leads the model to learn a *linear* pitch factor from unlabelled monophonic music audio in a self-supervised fashion. In addition, the same methodology can be applied to computer vision, learning a 3D Cartesian space from videos of a simple moving object without labels. Furthermore, physical symmetry naturally leads to *counterfactual representation augmentation*, a new technique which improves sample efficiency.

## 1 Introduction

Interpretable representation-learning models have achieved great progress for various types of time-series data. Taking the *music* domain as an example, tailored models [Ji *et al.*, 2020] have been developed to learn pitch, timbre, melody contour, chord progression, texture, etc. from music audio. These human-interpretable representations have greatly improved the performance of generative algorithms in various music creation tasks, including inpainting [Wei *et al.*, 2022], harmonization [Yi *et al.*, 2022], (re-)arrangement, and performance rendering [Jeong *et al.*, 2019].

However, most representation learning models still rely heavily on domain-specific knowledge. For example, to use pitch scales or instrument labels for learning pitch and timbre representations [Luo *et al.*, 2020, 2019; Engel *et al.*, 2020; Lin *et al.*, 2021; Esling *et al.*, 2018] and to use chords and rhythm labels for learning higher-level representations [Akama, 2019; Yang *et al.*, 2019; Wang *et al.*, 2020; Wei and Xia, 2021]. Such an approach is presumably very different from human learning; even without formal music training, we see that many people can learn *pitch*, a fundamental music concept, simply from the experience of listening to music. Hence, it remains an open question how to learn interpretable pitch factor using inductive biases that are more general. In other words, *what general computational principle gives rise to the concept of pitch.*

We see a similar issue in other domains. For instance, various computer-vision models [McCarthy and Ahmed, 2020; Trevithick and Yang, 2021; Mescheder *et al.*, 2019; Riegler *et al.*, 2017] can learn

3D representations of human faces or a particular scene by using domain knowledge (e.g., labelling of meshes and voxels, 3D convolution, etc.) But when these domain setups are absent, it remains a non-trivial task to learn the 3D location of a simple moving object in a self-supervised fashion.

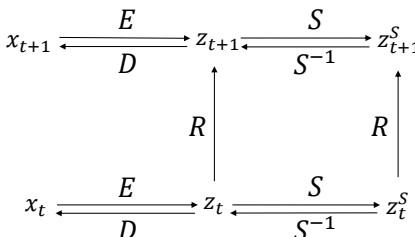

Figure 1: An illustration of physical symmetry as our inductive bias.

Inspired by modern physics, we explore to use *physical symmetry* (i.e., symmetry of physical laws) as a weak self-consistency constraint for the learned latent $z$ space of time-series data $x$. As indicated in Figure 1, this general inductive bias requires the learned prior model $R$, which is the induced physical law describing the temporal flow of the latent states, to be equivariant to a certain transformation $S$ (e.g., translation or rotation). Formally, $z_{t+1} = R(z_t)$ if and only if $z_{t+1}^S = R(z_t^S)$, where $z^S = S(z)$. In other words, $R$ and $S$ are commutable for $z$, i.e., $R(S(z)) = S(R(z))$. Note that our equivariant assumption applies only to the latent $z$ space. This is fundamentally different from most existing symmetry-informed models [Bronstein *et al.*, 2021], in which the equivariant property also imposes assumptions on the raw data space.

Specifically, we design **s**elf-supervised learning with **p**hysical **s**ymmetry (**SPS**)[1], a method that adopts an encoder-decoder framework and applies physical symmetry to the prior model. We show that SPS learns a *linear* pitch factor (that agree with human music perception) from monophonic music audio without any domain-specific knowledge about pitch scales, f0, or harmonic series. The same methodology can be applied to the computer vision domain, learning 3D Cartesian space from monocular videos of a bouncing ball shot from a fixed perspective. In particular, we see four desired properties of SPS as a self-supervised algorithm for interpretability:

- **Conciseness**: SPS does not require contrastive samples, bach normalization, or a large batch size. We can even drop the Gaussian prior regularization, which is usually required to learn a meaningful latent space. (See Section 3.2 - Section 4.)

- **Sample efficiency**: SPS learns low-dimensional representations of a dynamic system in a very sample-efficient way. (See Section 4 - Section 5.2.)

- **Robustness**: Even with an incorrect symmetry assumption, SPS can still learn more interpretable representations than baselines. (See Section 5.3.)

- **Extendability**: SPS can be easily combined with other learning techniques. For example, if we *further* assume an extra global invariant style code, the model becomes a disentangled sequential autoencoder, capable of learning content-style disentanglement from temporal signals. (See appendix.)

## 2    Intuition

The idea of using physical symmetry for representation learning comes from modern physics. In classical physics, scientists usually first induce physical laws from observations and then discover symmetry properties of the law. (E.g., Newton's law of gravitation, which was induced from planetary orbits, is symmetric with respect to Galilean transformations.) In contrast, in modern physics, scientists often start from a symmetry assumption, based on which they derive the corresponding law and predict the properties (representations) of fundamental particles. (E.g., general relativity was developed based on a firm assumption of symmetry with respect to Lorentz transformations).

---

[1]The source code is publicly available at https://github.com/XuanjieLiu/Self-supervised-learning-via-Physical-Symmetry. The demo page is available at https://xuanjieliu.github.io/SPS_demo/

Analogously, we use physical symmetry as an inductive bias of our representation learning model, which helps us learn a regularised prior and an interpretable low-dim latent space. If it is a belief of many physicists that symmetry in physical law is a major design principle of nature, we regard symmetry in physical law as a general inductive bias of perception. In other words, if physical symmetry leads an AI agent to learn human-aligned concepts in a self-supervised fashion, we believe that it could also provide insights into the ways that human minds perceive the world.

The introduction of physical symmetry naturally leads to **counterfactual representation augmentation**, a novel learning technique which helps improve sample efficiency. Representation augmentation means to "imagine" extra pairs of $z_t^S$ and $z_{t+1}^S$ as training samples for the prior model $R$. Through the lens of causality, this augmentation can be seen as a *counterfactual* inductive bias of the prediction on the representation level – *what if* the prior model makes predictions based on *transformed* latent codes? As indicated in Figure 1, it requires the prediction of the $z$ sequence to be *equivariant* to certain group transformations, $S$. This regularisation also constrains the encoder and decoder *indirectly through the prior model* since the network is trained in an end-to-end fashion.

## 3 Methodology

With physical symmetry, we aim to learn an interpretable low-dimensional representation $z_i$ of each high-dimensional sample $x_i$ from time-series $\mathbf{x}_{1:T}$. We focus on two problems in this paper: 1) to learn a *1D linear* pitch factor of music notes from music audio, where each $x_i$ is a spectrogram of a note, and 2) to learn *3D Cartesian location* factors of a simple moving object (a bouncing ball) from its trajectory shot by a fixed, single camera, where each $x_i$ is an image.

### 3.1 Model

Figure 2 shows the model design of SPS. During the training process, the temporal data input $\mathbf{x}_{1:T}$ is first fed into the encoder $E$ to obtain the corresponding representation $\mathbf{z}_{1:T}$. Then it is fed into *three* branches. In the first branch (the green line), $\mathbf{z}_{1:T}$ is decoded directly by the decoder $D$ to reconstruct $\mathbf{x}'_{1:T}$. In the second branch (the orange line), $\mathbf{z}_{1:T}$ is passed through the prior model $R$ to predict its next timestep, $\hat{\mathbf{z}}_{2:T+1}$, which is then decoded to reconstruct $\hat{\mathbf{x}}_{2:T+1}$. In the third branch (the blue line), we transform $\mathbf{z}_{1:T}$ with $S$, pass it through $R$, and transform it back using the inverse transformation $S^{-1}$ to predict another version of the next timestep $\tilde{\mathbf{z}}_{2:T+1}$, and finally decode it to $\tilde{\mathbf{x}}_{2:T+1}$. We get three outputs from the model: $\mathbf{x}'_{1:T}$, $\hat{\mathbf{x}}_{2:T+1}$, and $\tilde{\mathbf{x}}_{2:T+1}$.

The underlying idea of physical symmetry is that the dynamics of latent factor and its transformed version *follow the same physical law* characterised by $R$. Therefore, $\tilde{\mathbf{z}}$ and $\hat{\mathbf{z}}$ should be close to each other and so are $\tilde{\mathbf{x}}$ and $\hat{\mathbf{x}}$, assuming $S$ is a proper transformation. This self-consistency constraint helps the network learn a more regularised latent space.

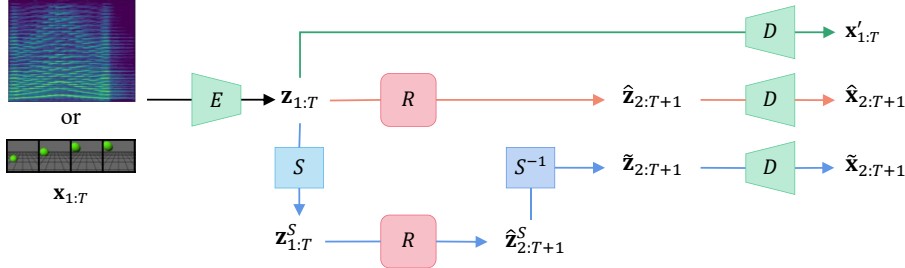

Figure 2: An overview of our model. $\mathbf{x}_{1:T}$ is fed into the encoder $E$ to obtain the corresponding representation $\mathbf{z}_{1:T}$, which is then fed into three different branches yielding three outputs respectively: $\mathbf{x}'_{1:T}$, $\hat{\mathbf{x}}_{2:T+1}$ and $\tilde{\mathbf{x}}_{2:T+1}$. Here, $R$ is the prior model and $S$ is the symmetric operation. The inductive bias of physical symmetry enforces $R$ to be equivaraint with respect to $S$, so $\tilde{\mathbf{z}}$ and $\hat{\mathbf{z}}$ should be close to each other and so are $\tilde{\mathbf{x}}$ and $\hat{\mathbf{x}}$.

## 3.2 Training objective

The total loss contains four terms: reconstruction loss $\mathcal{L}_{\text{rec}}$, prior prediction loss $\mathcal{L}_{\text{prior}}$, symmetry-based loss $\mathcal{L}_{\text{sym}}$, and KL divergence loss $\mathcal{L}_{\text{KLD}}$. Formally,

$$\mathcal{L} = \mathcal{L}_{\text{rec}} + \lambda_1 \mathcal{L}_{\text{prior}} + \lambda_2 \mathcal{L}_{\text{sym}} + \lambda_3 \mathcal{L}_{\text{KLD}}, \tag{1}$$

where $\lambda_1$, $\lambda_2$ and $\lambda_3$ are weighting parameters. By referring to the notations in section 3.1,

$$\mathcal{L}_{\text{rec}} = \mathcal{L}_{\text{BCE}}(\mathbf{x}'_{1:T}, \mathbf{x}_{1:T}) + \mathcal{L}_{\text{BCE}}(\hat{\mathbf{x}}_{2:T}, \mathbf{x}_{2:T}) + \mathcal{L}_{\text{BCE}}(\tilde{\mathbf{x}}_{2:T}, \mathbf{x}_{2:T}) \tag{2}$$

$$\mathcal{L}_{\text{prior}} = \ell_2(\hat{\mathbf{z}}_{2:T}, \mathbf{z}_{2:T}), \tag{3}$$

$$\mathcal{L}_{\text{sym}} = \ell_2(\tilde{\mathbf{z}}_{2:T}, \hat{\mathbf{z}}_{2:T}) + \ell_2(\tilde{\mathbf{z}}_{2:T}, \mathbf{z}_{2:T}). \tag{4}$$

$\mathcal{L}_{\text{KLD}}$ is the Kulback-Leibler divergence loss between the prior distribution of $z_i$ and a standard Gaussian. Lastly, we build two versions of SPS: SPS$_{\text{VAE}}$ and SPS$_{\text{AE}}$, with the latter replacing the VAE with an AE (and trivially doesn't have $\mathcal{L}_{\text{KLD}}$).

## 3.3 Symmetry-based counterfactual representation augmentation

During training, $S$ is the *counterfactual representation augmentation* since it creates extra imaginary sequences of $z$ (i.e., imaginary experience) to help train the prior. In practice, for each batch we apply $K$ different transformations $S_{1:K}$ to $\mathbf{z}$ and yield $K$ imaginary sequences. Thus, the two terms of symmetry-based loss can be specified as:

$$\ell_2(\tilde{\mathbf{z}}_{2:T}, \mathbf{z}_{2:T}) = \frac{1}{K} \sum_{k=1}^{K} \ell_2(S_k^{-1}(R(S_k(\mathbf{z}_{1:T-1}))), \mathbf{z}_{2:T}) \tag{5}$$

$$\ell_2(\tilde{\mathbf{z}}_{2:T}, \hat{\mathbf{z}}_{2:T}) = \frac{1}{K} \sum_{k=1}^{K} \ell_2(S_k^{-1}(R(S_k(\mathbf{z}_{1:T-1}))), \hat{\mathbf{z}}_{2:T}) \tag{6}$$

where the lower case $k$ denotes the index of a specific transformation and we refer to $K$ as the *augmentation factor*. Likewise, the last term of reconstruction loss can be specified as:

$$\mathcal{L}_{\text{BCE}}(\tilde{\mathbf{x}}_{2:T}, \mathbf{x}_{2:T}) = \frac{1}{K} \sum_{k=1}^{K} \mathcal{L}_{\text{BCE}}(D(S_k^{-1}(R(S_k(\mathbf{z}_{1:T-1})))), \mathbf{x}_{2:T}) \tag{7}$$

Each $S$ applied to each sequence $\mathbf{z}_{1:T}$ belongs to a certain group, and different groups are used for different problems. For the music problem, we assume $\mathbf{z}_i$ be to 1D and use random $S \in G \cong (\mathbb{R}, +)$. In other words, we add a random scalar to the latent codes. As for the video problem, we assume $\mathbf{z}_i$ be to 3D and use random $S \in G \cong (\mathbb{R}^2, +) \times \text{SO}(2)$. In other words, random rotation and translation are applied on two dimensions of $\mathbf{z}_i$.

## 4 Results

We test SPS under two modalities of temporal signals: music (section 4.1) and video (section 4.2). Each model is executed with 10 random initialisations, and evaluated on the test set. The highlight of this section is that SPS effectively learns interpretable low-dimensional factors that align with human perceptions. Also, by utilizing *small* training sets, we show the *high sampling efficiency* of our model. In the appendix, we further show that SPS also maintains accuracy in both reconstruction and prediction tasks (section A.2). Additionally, we present supplementary results trained on more complicated datasets and a more advanced configuration of our model, called SPS+, which enables content-style disentanglement in addition to interpretable learning. The findings from these more complex scenarios align closely with those observed in the simpler cases presented in this section.

### 4.1 Learning linear pitch factor from music audio

#### 4.1.1 Dataset and training setups

We synthesise a training dataset of 27 audio clips, each containing 15 notes in major scales with the first 8 notes ascending and the last 8 notes descending. We vary the starting pitch by integer numbers of MIDI pitch such that every MIDI pitch in the range A#4 to C7 is present in the training set. Only the accordion is used to generate the clips. For each clip in the training set, we uniformly randomly shift its pitch upwards by a decimal between 0 and 1, in this way generate the test set for evaluation.

We convert each audio clip into a sequence of image segments for processing. First, we run STFT (with sample rate $= 16000/s$, window length $= 1024$, hop length $= 512$, with no padding and no logarithmic frequency scale) over each audio clip to obtain a power spectrogram. After normalising the energy to the range $[0, 1]$, we slice the power spectrogram into fifteen image segments, each containing one note. The CNN encoder, in each timestep, takes one segment as input. For the latent space, we assume $\mathbf{z}_i \in \mathcal{R}$ and sample counterfactual representation augmentation $S \sim \mathcal{U}([-1, 1])$ where $S \in G \cong (\mathbb{R}, +)$. Note this assumption does not indicate any domain-specific inductive bias of music, such as the logarithmic relationship between pitch and frequency or the relationship between F0 and harmonics.

#### 4.1.2 Results on interpretable pitch space

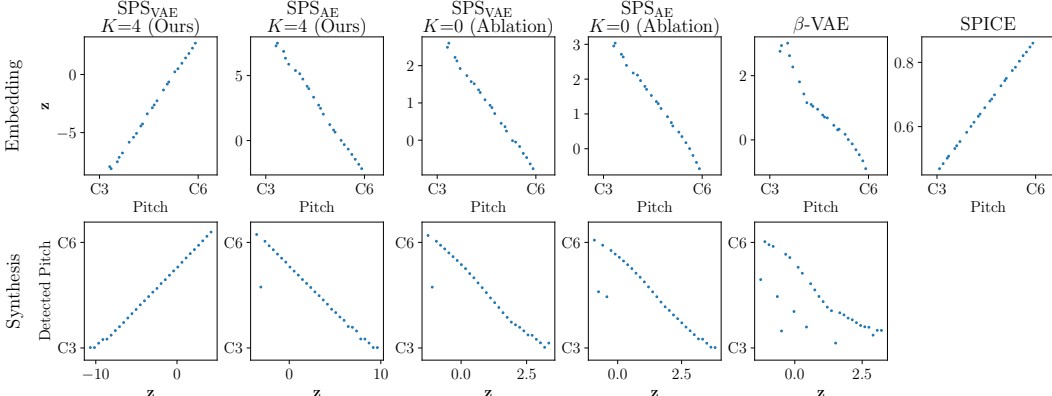

Figure 3: A visualisation of the mapping between the 1D learned factor $\mathbf{z}$ and the true pitch, in which a straight lines indicates a better result. In the upper row, models encode notes in the test set to $\mathbf{z}$. The $x$ axis shows the true pitch and the $y$ axis shows the learned pitch factor. In the lower row, the $x$ axis traverses the $\mathbf{z}$ space. The models decode $\mathbf{z}$ to audio clips. We apply YIN to the audio clips to detect the pitch, which is shown by the $y$ axis. In both rows, a linear, noiseless mapping is ideal, and our method performs the best.

Figure 3 demonstrates that the 1D pitch factor learned by our model exhibits a linear relationship with the conventional numerical ordering used to represent pitch by humans (e.g. MIDI pitch numbers). The plot shows the mappings of two tasks and six models. In the embedding task (the first row), the $x$-axis is the true pitch and the $y$-axis is embedded $\mathbf{z}$. In the synthesis task (the second row), the $x$-axis is $\mathbf{z}$ and the $y$-axis is the detected pitch (by YIN algorithm, a standard pitch-estimation method by [De Cheveigné and Kawahara, 2002]) of decoded (synthesised) notes. The first two models are SPS based on VAE and AE, respectively, trained with counterfactual representation augmentation factor $K = 4$. The third and fourth models are trained without constraints of physical symmetry ($K = 0$), serving as our ablations. The fifth one is a vanilla $\beta$-VAE, trained only to reconstruct, not to predict. The last one is SPICE [Gfeller *et al.*, 2020], a SOTA unsupervised pitch estimator *with strong domain knowledge on how pitch linearity is reflected in log-frequency spectrograms*. As the figure shows, 1) without explicit knowledge of pitch, our model learns a more interpretable pitch factor than $\beta$-VAE, and the result is comparable to SPICE, and 2) without the Gaussian prior assumption of latent variable distribution, our model SPS$_{AE}$ also learns a continuous representation space.

Table 1: The linearity of learned pitch factor and synthesized sound pitch evaluated by $R^2$.

| Methods | Learned factor $R^2 \uparrow$ | Synthesis $R^2 \uparrow$ |
|---|---|---|
| $\text{SPS}_\text{VAE}$, $K$=4 (Ours) | $0.999 \pm 0.001$ | $\mathbf{0.986 \pm 0.025}$ |
| $\text{SPS}_\text{AE}$, $K$=4 (Ours) | $0.998 \pm 0.001$ | $\mathbf{0.986 \pm 0.025}$ |
| $\text{SPS}_\text{VAE}$, $K$=0 (Ablation) | $0.997 \pm 0.002$ | $0.910 \pm 0.040$ |
| $\text{SPS}_\text{AE}$, $K$=0 (Ablation) | $0.993 \pm 0.006$ | $0.832 \pm 0.129$ |
| $\beta$-VAE | $0.772 \pm 0.333$ | $0.534 \pm 0.275$ |
| SPICE | $\mathbf{1.000}$ | N/A |

Table 1 shows a more quantitative analysis using $R^2$ as the metric to evaluate the linearity of the pitch against $z$ mapping from the encoder and the decoder. All models except SPICE are trained with 10 random initializations.

## 4.2 Learning object 3D coordinates from videos of a moving object

### 4.2.1 Dataset and training setups

We run physical simulations of a bouncing ball in a 3D space. The ball is randomly thrown and affected by gravity and the bouncing force (elastic force). A fixed camera records a 20-frame video of each 4-second simulation to obtain one trajectory (see Figure 4). The ball's size, gravity, and proportion of energy loss per bounce are constant across all trajectories. For each trajectory, the ball's initial location and initial velocity are randomly sampled. We utilize 512 trajectories for training, and an additional 512 trajectories for evaluation.

For the latent space, we set the dimension of the latent space to 3, but only constrain 2 of them by augmenting the representations with $S \in G \cong (\mathbb{R}^2, +) \times \text{SO}(2)$. Those two dimensions are intended to span the horizontal plane. The third one, which is the unaugmented latent dimension, is intended to encode the vertical height.

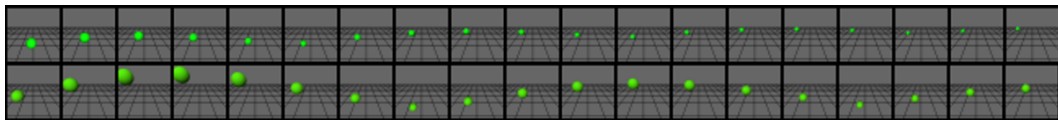

Figure 4: Two example trajectories from the bouncing ball dataset.

### 4.2.2 Results on interpretable 3D representation

Figure 5 visually evaluates the interpretability of the learned location factors by traversing the $z$ space, one dimension at a time, and using the learned decoder to synthesise images. If the learned factors are linear w.r.t. the 3D Cartesian coordinates, the synthesised ball should display linear motions as we change $z$ linearly. In brief, SPS learns an a more interpretable and linear $z$ space. Here, subplot (a) depicts the results of $\text{SPS}_\text{VAE}$ with $K$=4. We see that the un-augmented dimension, $z_2$, controls the height of the ball, while $z_1$ and $z_3$ move the ball along the horizontal (ground) plane. Each axis is much more linear than in (b) and (c). Subplot (b) evaluates $\text{SPS}_\text{VAE}$ with counterfactual representation augmentation $K$=0, essentially turning *off* SPS. As $z_i$ varies, the ball seems to travel along curves in the 3D space, showing the ablation learns some continuity w.r.t. the 3D space, but is obviously far from linear. In (c), the $\beta$-VAE fails to give consistent meaning to any axis.

Table 2 further shows quantitative evaluations on the linearity of the learned location factor, in which we see that SPS outperforms other models by a large margin. To measure linearity, we fit a linear regression from $z$ to the true 3D location over the test set and then compute the Mean Square Errors (MSE). Therefore, a smaller MSE indicates a better fit. To give an intuitive example, the MSEs of (a), (b) and (c) in Figure 5 are 0.09, 0.58 and 0.62 respectively. Here, we also include the results of $\text{SPS}_\text{AE}$. Very similar to the music experiment in session 4.1, we again see that even without the Gaussian prior assumption, our model $\text{SPS}_\text{AE}$ learns an interpretable latent space comparable to $\text{SPS}_\text{VAE}$.

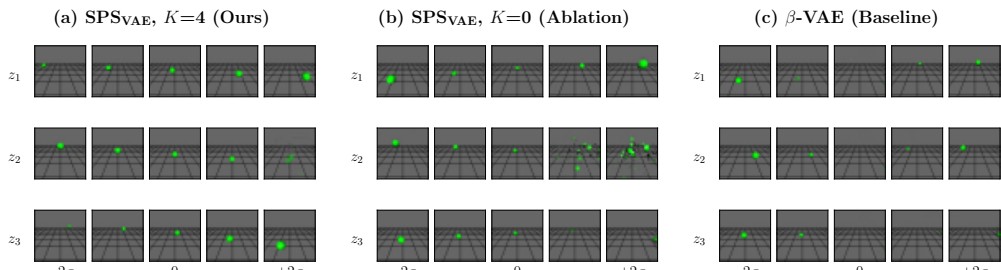

(a) SPS$_{\text{VAE}}$, $K$=4 (Ours) (b) SPS$_{\text{VAE}}$, $K$=0 (Ablation) (c) $\beta$-VAE (Baseline)

Figure 5: A visualisation of latent-space traversal performed on three models: (a) ours, (b) ablation, and (c) baseline, in which we see (a) achieves better linearity and interpretability. Here, row $i$ shows the generated images when changing $z_i$ and keeping $z_{\neq i} = 0$, where the $x$ axis varies $z_i$ from $-2\sigma_z$ to $+2\sigma_z$. We center and normalise $z$, so that the latent space from different runs is aligned for fair comparison. Specifically, in (a), changing $z_2$ controls the ball's height, and changing $z_1, z_3$ moves the ball parallel to the ground plane. In contrast, the behavior in (b) and (c) are less interpretable.

Table 2: Linear fits between the true location and the learned location factor. We run the encoder on the test set to obtain data pairs in the form of (location factor, true coordinates). We then run a linear fit on the data pairs to evaluate factor interpretability. Two outliers are removed from the 50 runs.

| Method | $x$ axis MSE $\downarrow$ | $y$ axis MSE $\downarrow$ | $z$ axis MSE $\downarrow$ | MSE $\downarrow$ |
|---|---|---|---|---|
| SPS$_{\text{VAE}}$, $K$=4 (Ours) | **0.11 $\pm$ 0.09** | **0.31 $\pm$ 0.34** | 0.26 $\pm$ 0.34 | 0.26 $\pm$ 0.30 |
| SPS$_{\text{AE}}$, $K$=4 (Ours) | 0.13 $\pm$ 0.07 | 0.39 $\pm$ 0.33 | **0.21 $\pm$ 0.17** | **0.24 $\pm$ 0.17** |
| SPS$_{\text{VAE}}$, $K$=0 (Ablation) | 0.33 $\pm$ 0.10 | 0.80 $\pm$ 0.18 | 0.75 $\pm$ 0.17 | 0.62 $\pm$ 0.14 |
| SPS$_{\text{AE}}$, $K$=0 (Ablation) | 0.26 $\pm$ 0.09 | 0.44 $\pm$ 0.27 | 0.55 $\pm$ 0.17 | 0.42 $\pm$ 0.15 |
| $\beta$-VAE | 0.36 $\pm$ 0.03 | 0.70 $\pm$ 0.01 | 0.68 $\pm$ 0.03 | 0.58 $\pm$ 0.01 |

## 5   Analysis

To better understand the effects of counterfactual representation augmentation (first introduced in section 3.3), we ran extra experiments with different $S$ and $K$. We choose the vision problem since a 3D latent space manifests a more obvious difference when physical symmetry is applied. In section 5.1, we show that a larger augmentation factor $K$ leads to higher sample efficiency. In section 5.2, we visualise the change of learned latent space against training epoch according to different values of $K$. In section 5.3, we show that some deliberately incorrect group assumptions $S$ can also achieve good results.

### 5.1   Counterfactual representation augmentation improves sample efficiency

Figure 6 shows that *a larger factor of counterfactual representation augmentation leads to a lower linear projection loss* (the measurement defined in section 4.2.2) of the learned 3D representation. Here, $K$ is the augmentation factor, and $K = 0$ means the model is trained without physical symmetry. The comparative study is conducted on 4 training set sizes (256, 512, 1024, and 2048), in which each box plot shows the results of 10 experiments trained with a fixed $K$ and random initialisation. We see that a larger $K$ leads to better results and compensates for the lack of training data. E.g., the loss trained on 256 samples with $K = 4$ is comparable to the loss trained on 1024 samples with $K = 0$, and the loss trained on 512 samples with $K = 4$ is even lower than the loss trained on 2048 samples with $K = 0$. Furthermore, when $K = 0$, increasing the number of training samples beyond a certain point does not further shrink the error, but increasing $K$ still helps.

### 5.2   Counterfactual representation augmentation improves interpretability

Figure 7 visualises the latent space during different stages of model training, and we see that a larger $K$ leads to a better enforcement of interpretability. The horizontal axis shows the training epoch. Three experiments with different $K$ values ($\times 0$, $\times 4$, $\times 16$) are stacked vertically. Each experiment is

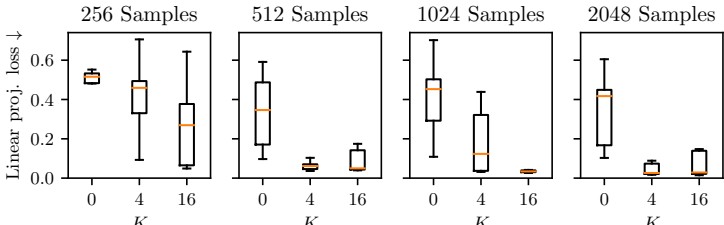

Figure 6: A comparison of linear projection MSEs among different augmentation factors ($K$) and training set sizes, which shows that counterfactual representation augmentation improves sample efficiency.

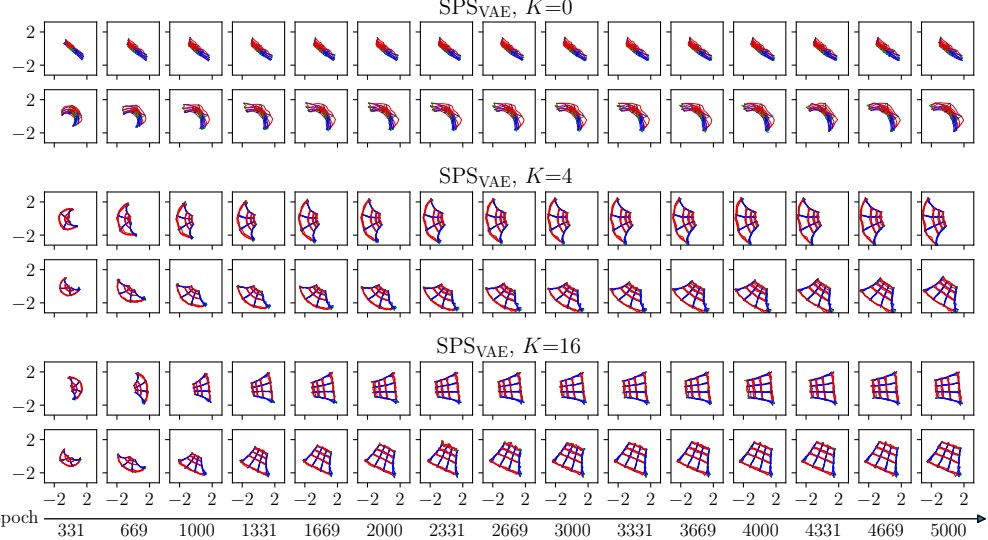

Figure 7: A visualisation of the learned latent space against training epoch, in which we see that a larger $K$ leads to a stronger enforcement on learning a linear latent space. Here, we plot how the encoder projects an equidistant 3D grid of true Cartesian coordinates onto the $z$ space. Different colours denote respective axes in the true coordinates.

trained twice with random initialisation. Each subplot shows the orthogonal projection of the $z$ space onto the plane spanned by $z_1$ and $z_3$, therefore hiding most of the $y$-axis (i.e. ball height) wherever a linear disentanglement is fulfilled. During training, the role of physical symmetry is to "straighten" the encoded grid and a larger $K$ yields a stronger effect.

## 5.3 Counterfactual representation augmentation with deliberately incorrect group assumptions

Additionally, we test SPS with deliberately incorrect group assumptions. The motivation is as follows. In real applications, researchers may incorrectly specify the symmetry constraint when the data are complex or the symmetry is not known *a priori*. SPS is more useful if it works with various groups assumptions close to the truth. In our analysis, we are surprised to find that SPS still learns interpretable representations under alternate group assumptions via perturbing the correct one.

Figure 8 shows our results with the vision task (on the bouncing ball dataset). The $x$ tick labels show the augmentation method. Its syntax follows section 3.3, e.g., "$(\mathbb{R}^1, +) \times SO(2)$" denotes augmenting representations by 1D translations and 2D rotations. The $y$ axis of the plot is still linear projection loss (as discussed in section A.5.3) that evaluates the interpretability of the learned representation. As is shown by the boxplot, five out of five perturbed group assumptions yield better results than the "w/o Symmetry" baseline. Particularly, $(\mathbb{R}^3, +) \times SO(2)$ and $(\mathbb{R}^2, +) \times SO(3)$ learn significantly more linear representations, showing that some symmetry assumptions are "less incorrect" than others, and that SPS can achieve good results under a multitude of group assumptions.

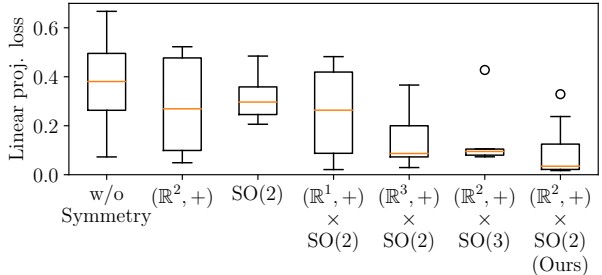

Figure 8: Evaluation on various group assumptions, which shows that physical symmetry is a very robust inductive bias as even the incorrect symmetry assumptions lead to better results than baseline. Here, the $y$ axis is linear projection loss between the learned location factor and the true coordinates, so a lower value means better interpretability of representations. The leftmost box shows the baseline without symmetry constraint. The next five boxes show five deliberately *incorrect* group assumptions, trained with $K$=4. The rightmost box shows the correct group assumption.

# 6 Related work

The idea of using a predictive model for better self-supervised learning has been well established [Oord *et al.*, 2018; Chung *et al.*, 2015; LeCun, 2022]. In terms of model architecture, our model is very similar to VRNN [Chung *et al.*, 2015]. In addition, our model can be seen as a variation of joint-embedding predictive architecture (JEPA) in [LeCun, 2022] if we eliminate the reconstruction losses on the observation. In fact, we see the network topology of a model as the "hardware" and see the learning strategy (e.g., contrastive method, regularised method, or a mixed one) as the "software". The main contribution of this study lies in the learning strategy — to use physical symmetry to limit the complexity of the prior model, and to use counterfactual representation augmentation to increase sample efficiency.

The existing notation of "symmetry" as in [Higgins *et al.*, 2018; Bronstein *et al.*, 2021] is very different from physical symmetry as an inductive for representation learning . Most current symmetry-based methods care about the relation between observation $x$ and latent $z$ [Sanghi, 2020; Quessard *et al.*, 2020; Dupont *et al.*, 2020; Huang *et al.*, 2021]. E.g., when a certain transformation is applied to $x$, $z$ should simply keep invariant or follow a same/similar transformation. Such an assumption inevitably requires some knowledge in the domain of $x$. In contrast, physical symmetry focuses solely on the dynamics of $z$, and therefore we only have to make assumptions about the underlying group transformation in the latent space. We see two most relevant works in the field of reinforcement learning [Mondal *et al.*, 2022; Dupont *et al.*, 2020], which apply an equivariant assumption similar to the physical symmetry used in this paper. The major differences are twofold. First, to disentangle the basic factors, our method requires no interactions with the environment. Second, our method is much more concise; it needs no other tailored components or other inductive biases such as symmetric embeddings network and contrastive loss used in [Dupont *et al.*, 2020] or MDP homomorphism applied in [Mondal *et al.*, 2022].

# 7 Limitation

We have identified several limitations in the generality and soundness of SPS. Firstly, when the underlying concept following physical symmetry only contains partial information of the time series and cannot fully reconstruct the inputs, SPS may not function properly. We hypothesize that this issue is connected to content-style disentanglement, and present some preliminary results in appendix A.4 and A.5. Secondly, the current model lacks the ability to distill concepts from multibody systems. For example, it is unable to learn the concept of pitch from polyphonic music or understand 3D space from videos featuring multiple moving objects. Lastly, it is essential to develop a formalized theory for quantifying the impact of counterfactual representation augmentation in future work. This would involve measuring the degree of freedom in the latent space with and without physical symmetry, and explaining why incorrect symmetry assumptions can still result in a correct and interpretable latent space.

## 8 Conclusion

In this paper, we use physical symmetry as a novel inductive bias to learn interpretable and low-dimensional representations from time-series data. Experiments show that physical symmetry effectively distills an interpretable linear pitch concept, which agrees with human music perception, from music audios without any labels. With the same method, we can learn the concept of 3D Cartesian space from monocular videos of bouncing ball shot from a fixed perspective. In addition, a robust training technique, counterfactual representation augmentation, is developed to enforce physical symmetry during training. Analysis shows that counterfactual representation augmentation leads to higher sample efficiency and better latent-space interpretability, and it stays effective even when the symmetry assumption is incorrect. Last but not least, we see that with physical symmetry, our sequential representation learning model can drop the the Gaussian prior regulation on the latent space. Such a result empirically indicates that physical symmetry, as a causal (counterfactual) inductive bias, might be more essential compared to the Gaussian prior as a purely statistical regularization.

**Acknowledgments**

We'd like to thank to Dr. Zhao Yang and Dr. Maigo Wang for inspiring discussion on physical symmetry. Also, we'd like to like to extend our sincere thanks to Junyan Jiang and Yixiao Zhang for their useful and practical suggestions.

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

# A   Appendix

The appendix is structured into three main parts.

The first part (section A.1, A.2) provides additional details about SPS. Section A.1 focuses on implementation-related aspects, while section A.2 presents experimental results concerning the reconstruction and prediction loss.

The second part (section A.3 to A.5) introduces an extended version of SPS called SPS+. Section A.3 describes the capabilities of SPS+ in achieving content-style disentanglement along with interpretable learning. The related experiments are presented in section A.4 and section A.5.

The third part (section A.6) presents two additional complex experiments conducted separately using SPS+ and SPS, respectively.

## A.1   SPS implementation details

### A.1.1   Architecture details

Our models for both tasks share the following architecture. The encoder first uses a 2D-CNN with ReLU activation to shrink the input down to an $8 \times 8$ middle layer, and then a linear layer to obtain $z$. If the encoder is in a VAE (instead of an AE), two linear layers characterises the posterior, one for the mean and the other for the log-variance. The prior model is a vanilla RNN of one layer with 256 hidden units and one linear layer projection head. The decoder consists of a small fully-connected network followed by 2D transposed convolution layers mirroring the CNN in the encoder. Its output is then passed through a sigmoid function. We use no batch normalisation or dropout layers.

Minor variations exist between the models for the two tasks. In the audio task, we use three convolution layers in the encoder, with three linear and three 2D transposed convolution layers in the decoder. In the vision task, as the data are more complex, we use four convolution layers in the encoder, with four linear and four 2D transposed convolution layers in the decoder.

### A.1.2   Training details

For both tasks, we use the Adam optimiser with learning rate $= 10^{-3}$. The training batch size is 32 across all of our experiments. For all VAE-based models, including $\text{SPS}_{\text{VAE}}$ (ours/ablation) and $\beta$-VAE (baseline), we set $\beta$ (i.e., $\lambda_3$ in Equation (1)) to 0.01, with $\lambda_1 = 1$ and $\lambda_2 = 2$. All BCE and MSE loss functions are calculated in sum instead of mean. $K = 4$ for all SPS models except for those discussed in section 5 where we analyse the influence of different $K$.

The RNN predicts $z_{n+1:T}$ given the first $n$ embeddings $z_{1:n}$. We choose $n = 3$ for the audio task and $n = 5$ for the vision task. We adopt scheduled sampling [Bengio *et al.*, 2015] during the training stage, where we gradually reduce the guidance from teacher forcing. After around 50000 batch iterations, the RNN relies solely on the given $z_{1:T}$ and predicts auto-regressively.

## A.2   SPS reconstruction and prior prediction results

We investigate the reconstruction and prediction capacities of our model and show that they are not harmed by adding symmetry constraints. For the music task, we compare our model, our model ablating symmetry constraints, and a $\beta$-VAE trained solely for the reconstruction of power spectrogram. Table 3 reports per-pixel BCE of the reconstructed sequences from the original input frames (Self-recon) and from the RNN predictions (Image-pred). We also include $\mathcal{L}_{\text{prior}}$, the MSE loss on the RNN-predicted $\hat{z}$ as defined in section 3.2. The results show that our models slightly surpasses the ablation and baseline models in all three metrics.

Similarly, Table 4 displays the reconstruction and prediction losses on the test set for the video task. Results show that adding symmetry constraints does not significantly hurt the prediction losses. Frame-wise self-reconstruction is significantly lower for the SPS models, but only by a small margin.

## A.3   SPS+

SPS can use physical symmetry to learn interpretable factors that evolve over time. We call those factors content representation. However, many problems can not be represented by content represen-

Table 3: Reconstruction and prediction results on the audio task.

| Methods | Self-recon ↓ | Image-pred ↓ | $\mathcal{L}_{\text{prior}}$ ↓ |
|---|---|---|---|
| SPS$_{\text{VAE}}$, $K$=4 (Ours) | 0.0292±0.0003 | 0.0296±0.0005 | **0.0012±0.0006** |
| SPS$_{\text{AE}}$, $K$=4 (Ours) | 0.0292±0.0002 | 0.0296±0.0002 | **0.0012±0.0003** |
| SPS$_{\text{VAE}}$, $K$=0 (Ablation) | **0.0291±0.0003** | **0.0295±0.0004** | 0.0030±0.0033 |
| SPS$_{\text{AE}}$, $K$=0 (Ablation) | **0.0291±0.0002** | **0.0295±0.0004** | 0.0087±0.0212 |
| $\beta$-VAE | 0.0303±0.0008 | N/A | N/A |

Table 4: Reconstruction and prediction losses of the video task. Two outliers are removed from the 50 runs.

| Method | Self-recon ↓ | Image-pred ↓ | $\mathcal{L}_{\text{prior}}$ ↓ |
|---|---|---|---|
| SPS$_{\text{VAE}}$, $K$=4 (Ours) | $0.64382 \pm 9e\text{-}05$ | $0.6456 \pm 4e\text{-}04$ | $0.14 \pm 0.05$ |
| SPS$_{\text{AE}}$, $K$=4 (Ours) | $0.64386 \pm 7e\text{-}05$ | $0.6458 \pm 3e\text{-}04$ | $0.17 \pm 0.07$ |
| SPS$_{\text{VAE}}$, $K$=0 (Ablation) | $0.64372 \pm 4e\text{-}05$ | $0.6459 \pm 2e\text{-}04$ | $0.19 \pm 0.10$ |
| SPS$_{\text{AE}}$, $K$=0 (Ablation) | $0.64367 \pm 5e\text{-}05$ | **$0.6456 \pm 1e\text{-}04$** | **$0.11 \pm 0.03$** |
| $\beta$-VAE | **$0.64345 \pm 5e\text{-}05$** | N/A | N/A |

tation alone. For example, the bouncing balls can have different colours and the pitch scales can be generated by different instruments. If the colour of a ball or the timbre of a sound scale are constant within a trajectory, those latent spaces are hard to constrain by physical symmetry. We call such invariant factors style representation. In order to deal with these problems, we combine SPS with a simple content-style disentanglement technique: SPS+, a more general framework of SPS. We use random pooling to constrain the style factors, and use physical symmetry to constrain the content representation in the same way as SPS in Section 3.1

### A.3.1 Model

Figure 9 shows the design of SPS+, which belongs to the family of disentangled sequential autoencoders [Bai *et al.*, 2021; Hsu *et al.*, 2017; Vowels *et al.*, 2021; Yingzhen and Mandt, 2018; Zhu *et al.*, 2020]. During the training process, the temporal data input $\mathbf{x}_{1:T}$ is first fed into the encoder $E$ to obtain the corresponding representation $\mathbf{z}_{1:T}$. $\mathbf{z}_{1:T}$ is then split into two parts: the style factor $\mathbf{z}_{1:T,s}$ and the content factor $\mathbf{z}_{1:T,c}$. The style factor $\mathbf{z}_{1:T,s}$ is passed through the random-pooling module $P$, where one element $z_{\tau,s}$ is randomly picked. The content factor $\mathbf{z}_{1:T,c}$ is fed into *three* branches, then combined with $z_{\tau,s}$ to reconstruct. For random pooling in the training stage, one style vector is randomly selected from all time steps (i.e., 15 for the music task and 20 for the vision task) of the sequence to represent $z_s$. In the testing stage, only the first 5 (vision task) or 3 (music task) frames are given, and $z_s$ will be selected from them.

### A.3.2 Training objective

The following loss functions in SPS+ slightly vary from those in SPS. For SPS+, $\mathcal{L}_{\text{prior}}$ and $\mathcal{L}_{\text{sym}}$ work on the content part of latent variables only. Other loss functions are exactly the same as those defined in section 3.2

$$\mathcal{L}_{\text{prior}} = \ell_2(\hat{\mathbf{z}}_{2:T,c}, \mathbf{z}_{2:T,c}), \tag{8}$$

$$\mathcal{L}_{\text{sym}} = \ell_2(\tilde{\mathbf{z}}_{2:T,c}, \hat{\mathbf{z}}_{2:T,c}) + \ell_2(\tilde{\mathbf{z}}_{2:T,c}, \mathbf{z}_{2:T,c}). \tag{9}$$

$$\ell_2(\tilde{\mathbf{z}}_{2:T,c}, \mathbf{z}_{2:T,c}) = \frac{1}{K} \sum_{k=1}^{K} \ell_2(S_k^{-1}(R(S_k(\mathbf{z}_{1:T-1,c}))), \mathbf{z}_{2:T,c}), \tag{10}$$

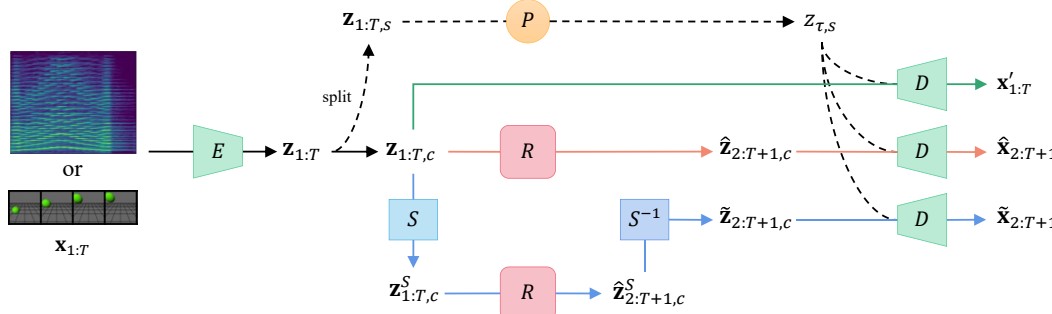

Figure 9: An overview of our model. $\mathbf{x}_{1:T}$ is fed into the encoder $E$ to obtain the corresponding representation $\mathbf{z}_{1:T}$, which is then split into two parts: the style factor $\mathbf{z}_{1:T,s}$ and the content factor $\mathbf{z}_{1:T,c}$. The style factor is passed through the random-pooling layer $P$, where an element $z_{\tau,s}$ is randomly selected. The content factor is fed into three different branches and combined with $z_{\tau,s}$ to reconstruct three outputs respectively: $\mathbf{x}'_{1:T}$, $\hat{\mathbf{x}}_{2:T+1}$ and $\tilde{\mathbf{x}}_{2:T+1}$. Here, $R$ is the prior model and $S$ is the symmetric operation. The inductive bias of physical symmetry enforces $R$ to be equivaraint w.r.t. to $S$, so $\tilde{\mathbf{z}}$ and $\hat{\mathbf{z}}$ should be close to each other and so are $\tilde{\mathbf{x}}$ and $\hat{\mathbf{x}}$.

$$\ell_2(\tilde{\mathbf{z}}_{2:T,c}, \hat{\mathbf{z}}_{2:T,c}) = \frac{1}{K}\sum_{k=1}^{K} \ell_2(S_k^{-1}(R(S_k(\mathbf{z}_{1:T-1,c}))), \hat{\mathbf{z}}_{2:T,c}), \tag{11}$$

$$\mathcal{L}_{\text{BCE}}(\tilde{\mathbf{x}}_{2:T}, \mathbf{x}_{2:T}) = \frac{1}{K}\sum_{k=1}^{K} \mathcal{L}_{\text{BCE}}(D(S_k^{-1}(R(S_k(\mathbf{z}_{1:T-1,c}))), z_{\tau,s}), \mathbf{x}_{2:T}). \tag{12}$$

## A.4   SPS+ on learning pitch & timber factors from audios of multiple instruments

### A.4.1   Dataset and setups

We synthesise a dataset that contains around 2400 audio clips played by **multiple instruments**. Similar to the dataset in section 4.1.1, each clip contains 15 notes in major scales with the first 8 notes ascending and the last 8 notes descending. Each note has the same volume and duration. The interval between every two notes is equal. We vary the starting pitch such that every MIDI pitch in the range C2 to C7 is present in the dataset. For each note sequence, we synthesise it using 53 different instruments, yielding 2376 audio clips. Specifically, two soundfonts are used to render those audio clips respectively: FluidR3_GM [Wen, 2013] for the train set and GeneralUser GS v1.471 [Chris, 2017] for the test set. The pitch ranges for different instruments vary, so we limit each instrument to its common pitch range (See Table 14).

We assume $z_c \in \mathcal{R}$ and $z_s \in \mathcal{R}^2$, and use random $S \in G \cong (\mathbb{R}, +)$ to augment $z_c$ with $K$=4.

### A.4.2   Results on pitch-timbre disentanglement

We evaluate the content-style disentanglement using factor-wise data augmentation following [Yang *et al.*, 2019]. Namely, we change (i.e., augment) the instrument (i.e., style) of notes while keeping their pitch the same, and then measure the effects on the encoded $z_c$ and $z_s$. We compare the normalised $z_c$ and $z_s$, ensuring they have the same dynamic range. Ideally, the change of $z_s$ should be much more significant than $z_c$. Here, we compare four approaches: 1) our model (SPS+), 2) our model without splitting for $z_s$ (SPS with $z \in \mathcal{R}^3$ and $S \in G \cong (\mathbb{R}, +)$) as an ablation, 3) GMVAE [Luo *et al.*, 2019], a domain-specific pitch-timbre disentanglement model trained with *explicit pitch labels*, and 4) TS-DSAE [Luo *et al.*, 2022], a recent unsupervised pitch-timbre disentanglement model based on Disentangled Sequential Autoencoder (DSAE).

Figure 10 presents the changes in normalised $z_c$ and $z_s$ measured by L2 distance when we change the instrument of an anchor note whose pitch is D3 and synthesised by accordion. Table 5 provides a more quantitative version by aggregating all possible instrument combinations and all different

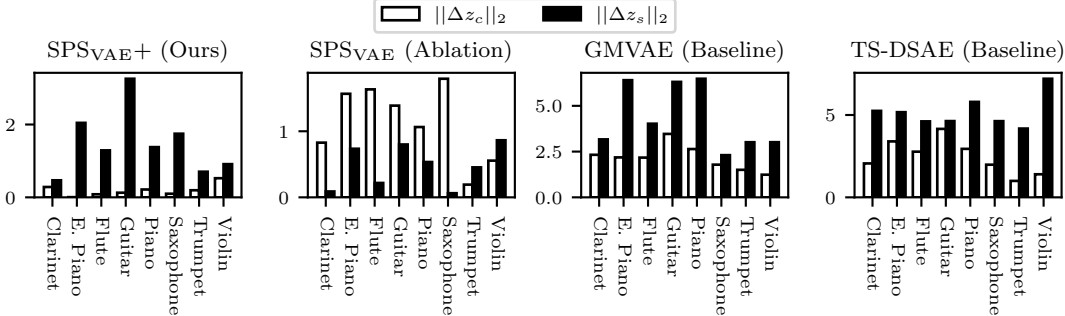

Figure 10: Comparisons for $\Delta z_c$ and $\Delta z_s$ for different instruments against accordion, with pitch kept constant at MIDI pitch D3. $\Delta z_c$ and $\Delta z_s$ are changes in normalised $z_c$ and $z_s$, so that higher black bars relative to white bars means better results. All results are evaluated on the test set.

Table 5: Mean ratios of changes in normalised $z_c$ and $z_s$ under timbre augmentation across all possible instrument combinations under different constant pitches in the test set.

| Methods | $||\Delta z_c||_2/||\Delta z_s||_2 \downarrow$ |
|---|---|
| SPS$_{VAE}$+ (Ours) | **0.49** |
| SPS$_{VAE}$  (Ablation) | 2.20 |
| GMVAE (Baseline) | 0.67 |
| TS-DSAE (Baseline) | 0.65 |

pitch pairs. Both results show that SPS+ produces a smaller relative change in $z_c$ under timbre augmentation, demonstrating a successful pitch-timbre disentanglement outperforming both the ablation and baseline. Note that for the ablation model, $z_c$ varies heavily under timbre augmentation, seemingly containing timbre information. This result indicates that the design of an invariant style factor over the temporal flow is necessary to achieve good disentanglement.

We further quantify the results in the form of augmentation-based queries following [Yang *et al.*, 2019], regarding the intended split in $z$ as ground truth and the dimensions with the largest variances from factor-wise augmentation after normalisation as predictions. For example, under timbre augmentation under a given pitch for our model, if $z_1$ and $z_3$ are the two dimensions of $z$ that produce the largest variances after normalisation, we count one false positive ($z_1$), one false negative ($z_2$), and one true positive ($z_3$). The precision would be 0.67. Table 6 shows the precision scores of the four approaches against their corresponding random selection. The results are in line with our observation in the previous evaluation, with our model more likely to produce the largest changes in dimensions in $z_c$ under content augmentation and that in $z_s$ under style augmentation.

Table 6: Results on augmentation-based queries on the audio task. Precision, recall and F1 are the same since the number of predicted and ground-truth positives are identical. Note that random precisions for different approaches can be different as $z_c$ and $z_s$ are split differently.

| Methods | Timbre augmentation | | Pitch augmentation | |
|---|---|---|---|---|
| | Precision $\uparrow$ | Random | Precision $\uparrow$ | Random |
| SPS$_{VAE}$+ (Ours) | **0.98** | 0.67 | 0.82 | 0.33 |
| SPS$_{VAE}$  (Ablation) | 0.50 | 0.67 | 0.02 | 0.33 |
| GMVAE (Baseline) | 0.93 | 0.50 | **0.83** | 0.50 |
| TS-DSAE (Baseline) | 0.81 | 0.50 | 0.68 | 0.50 |

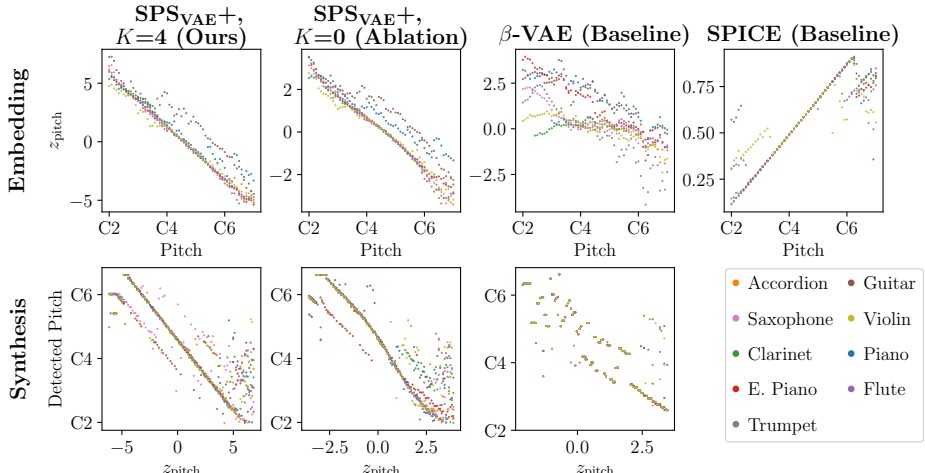

Figure 11: A visualisation of the mapping between the 1D content factor and the true pitch. In the upper row, models encode notes in the test set to $z_{\text{pitch}}$. The $x$ axis shows the true pitch and the $y$ axis shows the learned pitch factor. In the lower row, the $x$ axis traverses the $z_{\text{pitch}}$ space. The models decode $z_{\text{pitch}}$ to audio clips. We apply YIN to the audio clips to detect the pitch, which is shown by the $y$ axis. In both rows, a linear, noiseless mapping is ideal, and our method performs the best.

Table 7: Reconstruction and prediction results on the audio task.

| Methods | Self-recon ↓ | Image-pred ↓ | $\mathcal{L}_{\text{prior}}$ ↓ |
|---|---|---|---|
| SPS$_{\text{VAE}}$+, $K$=4 (Ours) | **0.0356** | **0.0359** | **0.0418** |
| SPS$_{\text{VAE}}$+, $K$=0 (Ablation) | 0.0360 | 0.0363 | 0.0486 |
| $\beta$-VAE (Baseline) | 0.0359 | N/A | N/A |

### A.4.3 Results on interpretable pitch space

Figure 11 shows that the pitch factor learned by SPS+ has a linear relation with the true pitch. Here, we use $z_{\text{pitch}}$ as the synonym of $z_c$ to denote the content factor. The plot shows the mappings of two tasks and four models. In the embedding task (the first row), $x$-axis is the true pitch and $y$-axis is embedded $z_{\text{pitch}}$. In the synthesis task (the second row), $x$-axis is $z_{\text{pitch}}$ and $y$-axis is the detected pitch (by YIN algorithm, a standard pitch-estimation method by [De Cheveigné and Kawahara, 2002]) of decoded (synthesised) notes. The fours models involved are: 1) our model, 2) our model without symmetry ($K$=0), 3) a $\beta$-VAE trained to encode single-note spectrograms from a single instrument (banjo) to 1D embeddings, and 4) SPICE [Gfeller *et al.*, 2020], a SOTA unsupervised pitch estimator *with strong domain knowledge on how pitch linearity is reflected in log-frequency spectrograms*. As the figure shows, without explicit knowledge of pitch, our model learns a more interpretable pitch factor than $\beta$-VAE, and the result is comparable to SPICE.

Figure 12 shows a more quantitative analysis, using $R^2$ as the metric to evaluate the linearity of the pitch against $z_{\text{pitch}}$ mapping. Although SPICE produces rather linear mappings in Figure 11, it suffers from octave errors towards extreme pitches, hurting its $R^2$ performance.

### A.4.4 Reconstruction and prior prediction

We investigate the reconstruction and prediction capacities of our model and show that they are not harmed by adding symmetry constraints. We compare our model, our model ablating symmetry constraints, and a $\beta$-VAE trained solely for only image reconstruction. Table 7 reports per-pixel BCE of the reconstructed sequences from the original input frames (Self-recon) and from the RNN predictions (Image-pred). We also include $\mathcal{L}_{\text{prior}}$, the MSE loss on the RNN-predicted $\hat{z}$ as redefined in section A.3.2. The results show that our model surpasses the ablation and baseline models in all three indexes.

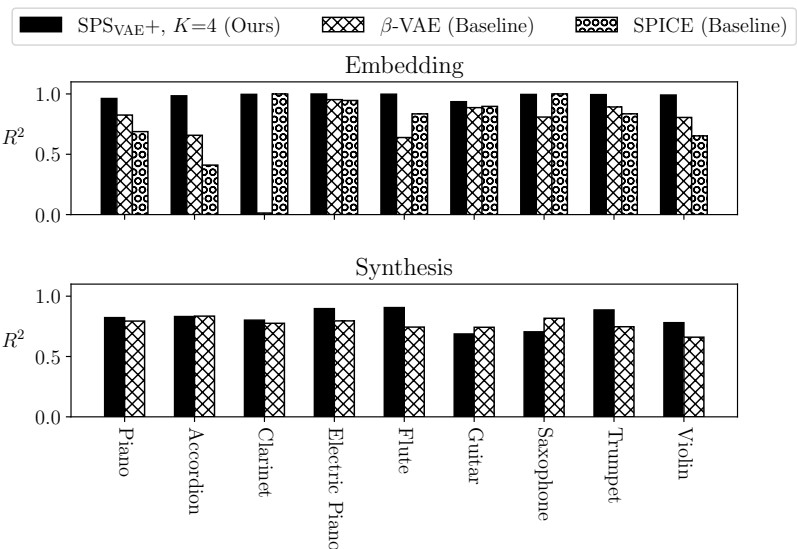

Figure 12: We use $R^2$ to evaluate mapping linearity. A larger $R^2$ indicates a more interpretable latent space. Results are evaluated on the test set.

## A.5 SPS+ on learning space & colour factors from videos of colourful bouncing balls

### A.5.1 Dataset and setups

We run physical simulations of a bouncing ball in a 3D space and generate 4096 trajectories, yielding a dataset of videos. Similar to the dataset in section 4.2.1, the simulated ball is affected by gravity and bouncing force (elastic force). A fixed camera records a 20-frame video of each 4-second simulation to obtain one trajectory (see Figure 13). The ball's size, gravity, and proportion of energy loss per bounce are constant across all trajectories. In this dataset, the color of the ball varies by trajectory, rather than a single color. For each trajectory, the ball's colours are uniformly randomly sampled from a continuous colour space.

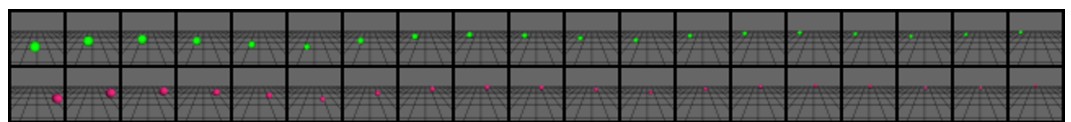

Figure 13: Two example trajectories from the bouncing ball dataset.

We set $z_c \in \mathcal{R}^3$ with the same counterfactual representation augmentation as in section 4.2 ($S \in G \cong (\mathbb{R}^2, +) \times \mathrm{SO}(2)$, $K$=4). Two of its dimensions are intended to span the horizontal plane and the third unaugmented latent dimension is intended to encode the vertical height. We set $z_s \in \mathcal{R}^2$ which is intended to represent the ball's colour space.

### A.5.2 Result on space-colour disentanglement

Similar to section A.4.2, we evaluate the space-colour disentanglement by augmenting the colour (i.e., style) of the bouncing balls while keeping their locations, and then measure the effects on the normalised $z_c$ and $z_s$. Again, a good disentanglement should lead to a change in $z_s$ much more significant than $z_c$. Here, we compare two approaches: 1) our model (SPS+) and 2) our model ablating splitting for $z_s$ (SPS with $z \in \mathcal{R}^5$ and $S \in G \cong (\mathbb{R}^2, +) \times \mathrm{SO}(2)$). Note that the ablation model does not differently constrain $z_2$ (corresponding to the $y$-axis) than $z_s$. To ensure a meaningful comparison, under colour augmentation, we consider $z_2$ to be a part of $z_s$ of the ablation model and a part of $z_c$ of the complete model.

Figure 14 presents the changes in normalised $z_c$ and $z_s$ measured by L2 distance when we change the colour of an anchor ball whose location is (0, 1, 5) and rendered using white colour. Table 9

Table 8: Results on augmentation-based queries on the visual task. Since the ablation model does not differently constrain $z_2$ (corresponding to the $y$-axis) than $z_s$, we consider $z_c$ and $z_s$ differently for the two approaches. Under colour augmentation, we consider $z_2$ to be a part of $z_s$ for the ablation model and a part of $z_c$ for the complete model. Under location augmentation, we consider $z_2$ to be a part of $z_c$ for both models.

| Methods | Colour augmentation | | Location augmentation | |
|---|---|---|---|---|
| | Precision ↑ | Random | Precision ↑ | Random |
| $SPS_{VAE}$+ (Ours) | **0.99** | 0.40 | **0.88** | 0.40 |
| $SPS_{VAE}$ (Ablation) | 0.64 | 0.60 | 0.36 | 0.40 |

Table 9: Mean ratios of changes in normalised $z_c$ and $z_s$ under colour augmentation across sampled colour combinations keeping locations constant. Results are evaluated on the test set.

| Methods | $||\Delta z_c||_2/||\Delta z_s||_2 \downarrow$ |
|---|---|
| $SPS_{VAE}$+ (Ours) | **0.54** |
| $SPS_{VAE}$ (Ablation) | 1.62 |

provides a more quantitative version by aggregating sampled colour combinations and location pairs. Both results show that our model produces a smaller relative change in $z_c$ under timbre augmentation, demonstrating a successful pitch-timbre disentanglement outperforming the ablation model. Note that for the ablation model, $z_c$ varies heavily under colour augmentation. Table 8 shows the precision scores of the SPS+ and its ablation against their corresponding random selection for the ball task. These results agree with section A.4.2 and again indicate that the design of an invariant style factor helps with disentanglement.

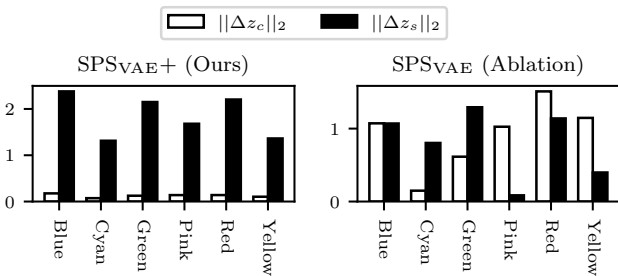

Figure 14: Comparisons of normalised $\Delta z_c$ and $\Delta z_s$ for different colours against white, with the ball's location kept constant at (0, 1, 5). Higher black bars (relative white bars) means better a result. (Results are evaluated on the test set.)

Figure 15 evaluates the learned colour factor of our model. Each pixel shows the colour of the ball synthesised by the decoder using different $z$ coordinates. The ball colour is detected using naive saturation maxima. In the central subplot, the location factor $z_{1:3}$ stays at zeros while the colour factor $z_{4:5}$ is controlled by the subplot's $x, y$ axes. As shown in the central subplot, our model (a) learns a natural 2D colour space. The surrounding subplots keep the colour factor $z_{4:5}$ unchanged, and the location factor $z_{1,3}$ is controlled by the subplot's $x, y$ axes. A black cross marks the point where the entire $z_{1:5}$ is equal to the corresponding black cross in the central subplot. As is shown by the surrounding subplots, varying the location factor does not affect the colour produced by our model (a), so the disentanglement is successful. The luminosity changes because the scene is lit by a point light source, making the ball location affect the surface shadow. On the other hand, $\beta$-VAE (b) learns an uninterpretable colour factor.

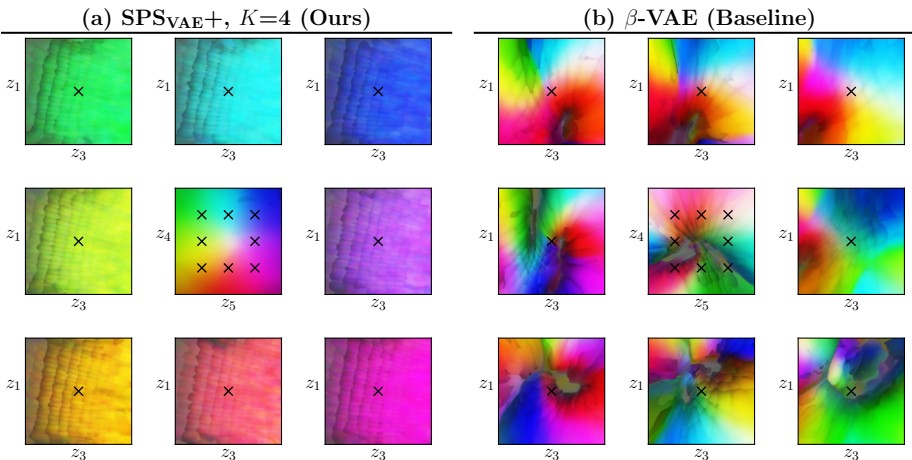

Figure 15: The colour map of the synthesised ball experiment through latent space traversal. Each pixel represents the detected colour from one synthesised image of the ball. Each subplot varies two dimensions of $z$, showing how the synthesised colour responds to the controlled $z$.

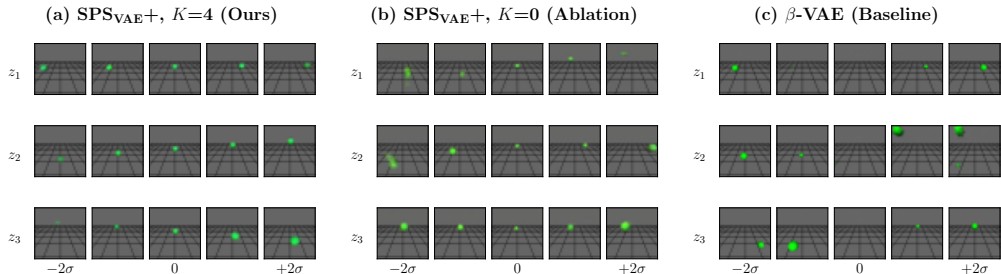

Figure 16: Row $i$ shows the generated images when changing $z_i$ and keeping $z_{\neq i} = 0$, where the $x$ axis varies $z_i$ from $-2\sigma$ to $+2\sigma$. In (a), changing $z_2$ controls the ball's height, and changing $z_1, z_3$ moves the ball parallel to the ground plane.

### A.5.3 Results on interpretable 3D representation

Figure 16 illustrates the interpretability of learned content factor using latent space traversal. Each row varies only one dimension of the learned 3D content factor, keeping the other two dimensions at zero. Figure 16(a) shows the results of our model. We clearly observe that: i) *increasing $z_1$ (the first dimension of $z_c$ ) mostly moves the ball from left to right, increasing $z_2$ moves the ball from bottom to top, and increasing $z_3$ mostly moves the ball from far to near.* Figure 16(b) is the ablation model without physical symmetry, and (c) shows the result of our baseline model $\beta$-VAE, which is trained to reconstruct static images of a single colour (green). Neither (b) nor (c) learns an interpretable latent space.

Table 10: Linear fits between the true location and the learned location factor. We run the encoder on the test set to obtain data pairs in the form of (location factor, true coordinates). We then run a linear fit on the data pairs to evaluate factor interpretability.

| Method | $x$ axis MSE $\downarrow$ | $y$ axis MSE $\downarrow$ | $z$ axis MSE $\downarrow$ | MSE $\downarrow$ |
|---|---|---|---|---|
| SPS$_{\text{VAE}}$+, $K$=4 (Ours) | **0.11** | **0.06** | **0.09** | **0.09** |
| SPS$_{\text{VAE}}$+, $K$=0 (Ablation) | 0.35 | 0.72 | 0.68 | 0.58 |
| $\beta$-VAE (Baseline) | 0.37 | 0.76 | 0.73 | 0.62 |

Table 11: Reconstruction and prediction results on the video task with variable colours.

| Method | Self-recon ↓ | Image-pred ↓ | $\mathcal{L}_{\mathrm{prior}}$ ↓ |
|---|---|---|---|
| SPS$_{\mathrm{VAE}}$+, $K$=4 (Ours) | 0.6457 | **0.6464** | **0.0957** |
| SPS$_{\mathrm{VAE}}$+, $K$=0 (Ablation) | 0.6456 | **0.6464** | 0.1320 |
| $\beta$-VAE (Baseline) | **0.6455** | N/A | N/A |

Table 12: $R^2$ aggregated across all instruments in the test set. A larger $R^2$ indicates a more interpretable latent space.

| Method | Self-recon ↓ | Image-pred ↓ | $\mathcal{L}_{\mathrm{prior}}$ ↓ | Embedding $R^2$ ↑ | Synthesis $R^2$ ↑ |
|---|---|---|---|---|---|
| SPS$_{\mathrm{VAE}}$+, $K$=4 (Ours) | 0.0384 | **0.0396** | **0.7828** | **0.89** | **0.47** |
| SPS$_{\mathrm{VAE}}$+, $K$=0 (Ablation) | 0.0388 | 0.0400 | 0.9909 | 0.83 | 0.25 |
| $\beta$-VAE (Baseline) | **0.0324** | N/A | N/A | 0.19 | 0.29 |

Table 10 quantitatively evaluates the linearity of the learned location factor. We fit a linear regression from $z_c$ to the true 3D location over the test set and then compute the Mean Square Errors (MSEs). A smaller MSE indicates a better fit. All three methods (as used in Figure 16) are evaluated on a single-colour (green) test set. Results show that our model achieves the best linearity in the learned latent factors, which aligns with our observations in Figure 16.

### A.5.4 Reconstruction and prior prediction

Similar to section A.4.4, we show that our model suffers little decrease in reconstruction and prediction performance while surpassing the ablation model in terms of $\mathcal{L}_{\mathrm{prior}}$ by table 11.

### A.6 More complicated tasks

The main part of this paper focuses on simple, straight-forward experiments. Still, we supplement our findings by reporting our current implementation's performance on more complicated tasks involving natural melody and real-world video data.

### A.6.1 Learning interpretable pitch factors from natural melodies

We report the performance of SPS+ on learning interpretable pitch factors from monophonic melodies under a more realistic setup. We utilize the melodies from the Nottingham Dataset [Foxley, 2011], a collection of 1200 American and British folk songs. For simplicity, we quantise the MIDI melodies by eighth notes, replace rests with sustains and break down sustains into individual notes. We synthesise each non-overlapping 4-bar segment with the accordion soundfonts in FluidR3 GM [Wen, 2013], resulting in around 5000 audio clips, each of 64 steps.

This task is more realistic than the audio task described in A.4 since we use a large set of natural melodies instead of one specified melody line. The task is also more challenging as the prior model has to predict long and more complex melodies. To account for this challenge, we use a GRU [Cho et al., 2014] with 2 layers of 512 hidden units as the prior model. We perform early-stopping after around 9000 iterations based on spectrogram reconstruction loss on the training set. The model and training setup is otherwise the same as in A.4.

Following A.4.3, We evaluate our approach on notes synthesised with all instruments in GeneralUser GS v1.471 [Chris, 2017] in the MIDI pitch range of C4 to C6, where most of the melodies in Foxley [2011] take place. Note that this is a challenging zero-shot scenario since the model is trained on only one instrument. We compare our model, our model ablating the symmetry loss and a $\beta$-VAE baseline. We visualise the embedded $z_{\mathrm{pitch}}$ and synthesised pitches for different instruments in Figure 17. Following 12, $R^2$ results are shown in Figure 18 and Table 12. Even when tested on unseen timbres, our model can learn linear and interpretable pitch factors and demonstrates better embedding and synthesis performance compared with the ablation model, which outperforms the $\beta$-VAE baseline.

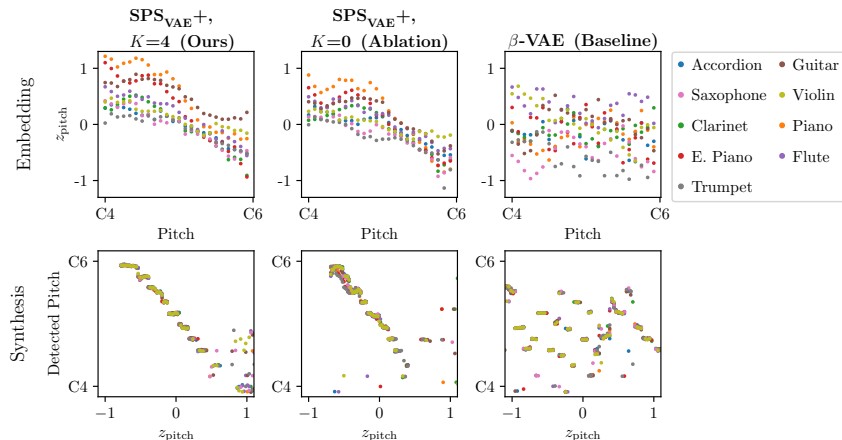

Figure 17: A visualisation of the mapping between the embedded 1D content factor and the true pitch for the model trained on Nottingham dataset.

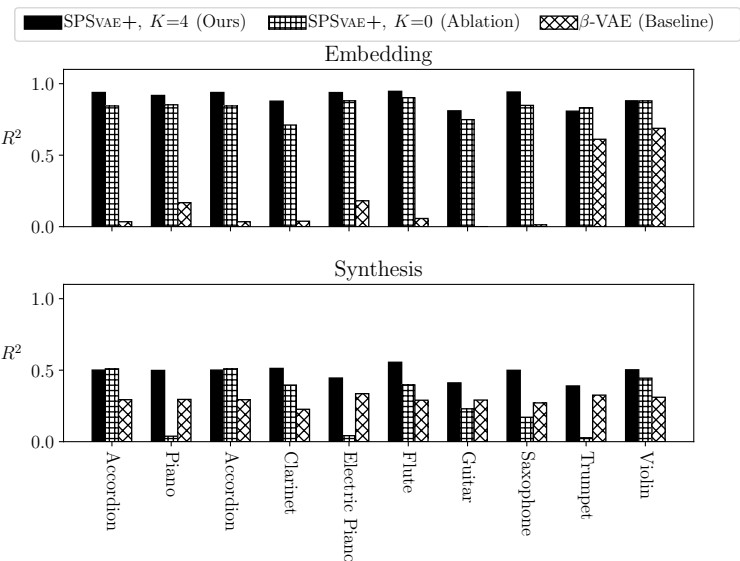

Figure 18: $R^2$ for select instruments in the test set. A larger $R^2$ indicates a more linear and interpretable latent space.

### A.6.2 Learning an intepretable location factor from KITTI-Masks

In this task, we evaluate our method's capability on a real-world dataset, KITTI-Masks [Klindt *et al.*, 2021]. The dataset provides three labels for each image: $X$ and $Y$ for the mask's 2D coordinate, and $AR$ for the pixel-wise area of the mask. Based on the provided labels, we use simple geometrical relation to estimate the person-to-camera distance $d$, computed as $d = 1/\tan(\alpha\sqrt{AR})$, where $\alpha$ is a constant describing the typical camera's Field of View (FoV).

We use a 3-dimensional latent code for all models. For SPS, all 3 dimensions are content factors $z_c$ and no style factor $z_s$ is used. We apply group assumption $(\mathbb{R}^3, +)$ to augment representations with $K = 1$. To measure the interpretability, we fit a linear regression from $z_c$ to the ground truth labels and calculate MSEs in the same way as in section A.5.3. The results are shown in Table 13. Linear proj. MSE 1 measures the errors of linear regression from $z_c$ to the original dataset labels. Linear proj. MSE 2 measures the errors of linear regression from $z_c$ to the person's 3-D location, estimated from the labels.

Table 13: Results of KITTI-Masks task, averaging on 30 random initialisations for each method.

| Methods | Self-recon ↓ | Image-pred ↓ | Linear proj. MSE 1 ↓ | Linear proj. MSE 2 ↓ |
|---|---|---|---|---|
| SPS$_{\text{VAE}}$, $K$=4 (Ours) | 0.030±0.001 | **0.084±0.006** | **0.215±0.067** | **0.203±0.065** |
| SPS$_{\text{VAE}}$, $K$=0 (Ablation) | 0.030±0.001 | 0.093±0.010 | 0.235±0.077 | 0.243±0.088 |
| $\beta$-VAE (Baseline) | **0.028±0.001** | N/A | 0.403±0.194 | 0.399±0.204 |

As is shown in Table 13, MSE 2 is smaller than MSE 1 for SPS, indicating that SPS learns more fundamental factors (person's location) rather than superficial features (pixel-wise location and area). For the baseline methods, MSE 2 is almost equal to MSE 1, and both of them are higher than those of SPS. In summary, our experiment shows that SPS learns more interpretable representations than the baseline (as well as the ablation method) on KITTI-Masks dataset.

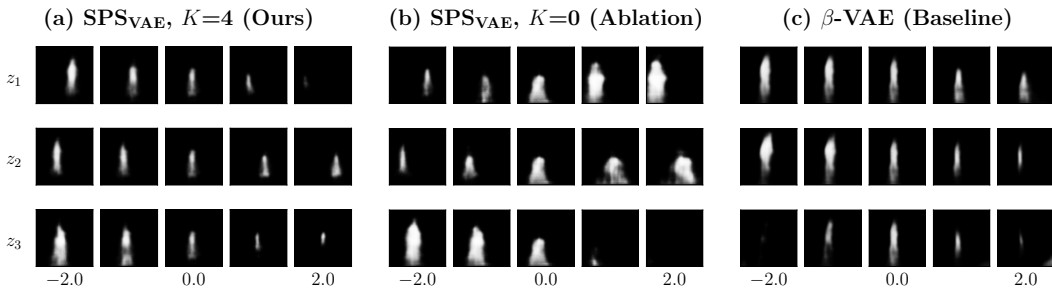

Figure 19: Latent space traversal on different models. Row $i$ shows the generated images when changing $z_i$ and keeping $z_{\neq i} = 0$. For our model, the range of $z_1$ from -2 to 2 corresponds to the human location from near-right to far-left, $z_2$ from near-left to far-right, and $z_3$ from near to far. We can see that other methods produce more non-linear trajectories, for example in (c), the human location hardly changes when $z_1 < 0$, but it changes dramatically when $z_1 > 0$.

Figure 19 shows the generated images, which illustrates that the factors learned by SPS are more linear than those learned by other methods in the human location attribute. For the experiment, We choose all sequences with length $\geq 12$ from KITTI-Masks as our dataset; we use 1058 sequences for training and 320 sequences for evaluation; In the inference stage, only the first 4 frames are given. All three methods are trained 30 times with different random initializations. Table 13 shows the average results evaluated on the same test set with 30 different seeds.

| Instrument | MIDI Note (from) | MIDI Note (to) |
|---|---|---|
| Accordion | 58 | 96 |
| Acoustic Bass | 48 | 96 |
| Banjo | 36 | 96 |
| Baritone Saxophone | 36 | 72 |
| Bassoon | 36 | 84 |
| Celesta | 36 | 96 |
| Church Bells | 36 | 96 |
| Clarinet | 41 | 84 |
| Clavichord | 36 | 84 |
| Dulcimer | 36 | 84 |
| Electric Bass | 40 | 84 |
| Electric Guitar | 36 | 96 |
| Electric Organ | 36 | 96 |
| Electric Piano | 36 | 96 |
| English Horn | 36 | 85 |
| Flute | 48 | 96 |
| Fretless Bass | 36 | 84 |
| Glockenspiel | 36 | 96 |
| Guitar | 36 | 96 |
| Harmonica | 36 | 96 |
| Harp | 36 | 96 |
| Harpsichord | 36 | 96 |
| Horn | 36 | 96 |
| Kalimba | 36 | 96 |
| Koto | 36 | 96 |
| Mandolin | 36 | 96 |
| Marimba | 36 | 96 |
| Oboe | 36 | 96 |
| Ocarina | 36 | 96 |
| Organ | 36 | 96 |
| Pan Flute | 36 | 96 |
| Piano | 36 | 96 |
| Piccolo | 48 | 96 |
| Recorder | 36 | 96 |
| Reed Organ | 36 | 96 |
| Sampler | 36 | 96 |
| Saxophone | 36 | 84 |
| Shakuhachi | 36 | 96 |
| Shamisen | 36 | 96 |
| Shehnai | 36 | 96 |
| Sitar | 36 | 96 |
| Soprano Saxophone | 36 | 96 |
| Steel Drum | 36 | 96 |
| Timpani | 36 | 96 |
| Trombone | 36 | 96 |
| Trumpet | 36 | 96 |
| Tuba | 36 | 72 |
| Vibraphone | 36 | 96 |
| Viola | 36 | 96 |
| Violin | 36 | 96 |
| Violoncello | 36 | 96 |
| Whistle | 48 | 96 |
| Xylophone | 36 | 96 |

Table 14: Pitch range (in MIDI note) for each instrument in our dataset.

