# OpenReview forum: "Learning Interpretable Low-dimensional Representation via Physical Symmetry"
_NeurIPS.cc/2023/Conference — NeurIPS 2023 poster_

### Official Review · Reviewer_TYBC · 2023-06-25

**Soundness:** 3 good
**Presentation:** 2 fair
**Contribution:** 3 good
**Rating:** 4
**Confidence:** 3

**Summary:**

This paper presents an approach to interpretable representation learning based on ''physical symmetry''. The core idea is to learn to predict the temporal evolution of a latent variable which is additionally encouraged to be equivariant under some transformation (e.g. translation or rotation). An autoencoder-like model is designed, where an input sequence is encoded to a sequence of latent variables. From these, three objectives are defined: direct reconstruction, reconstruction after predicting the next state, and an objective involving the symmetric transform. Experiments are presented on an audio-related task (predicting pitches from a sequence of single-note input spectrograms) and a vision task (predicting the coordinates of a moving ball). Further experiments are performed to demonstrate a reduced need for training examples, interpretability of the learned representations, and style-content disentanglement.

**Strengths:**

The main idea at the core of this work is interesting and original. This idea might also stimulate future work using such physical symmetry constraints.

The paper provides a variety of experiments, which seem sensibly designed. It is welcome to see experiments dedicated to two different domains (vision and audio). I especially liked the experiments on style-content disentanglement in the Appendix (SPS+). These contain much more realistic scenarios than the main text (albeit still on synthetic datasets).

The writing in the later sections of the paper is mostly clear and the figures throughout the manuscript are well done. More on the writing below.

The proposed approach shows only slight or moderate improvements compared to using no physical symmetry constraints. Nevertheless, the idea itself could be considered significant enough to be shared with the community.

**Weaknesses:**

In my view, the paper suffers from three main weaknesses:

- The writing in the initial sections of the paper (even the title!) is unnecessarily convoluted and confusing. Initially, I could not understand the main idea of the approach until reading the results sections and the appendix.
- The experiments are performed on very constrained, synthetic datasets. The train-test split of the audio experiment, especially, seems very simple (the test set contains the same sequences as the training set, just very slightly detuned). For example, based on these experiments, I do not believe the proposed model would work for pitch estimation in-the-wild (whereas SPICE does, to some extent).
- The results for the proposed approach show a slight or moderate improvement compared to the K=0 baseline. However, most of the observed effect seems to come from the sequence prediction and reconstruction objectives, and not from the proposed physical symmetry (see, e.g., Figure 3, K=0 vs K=4). This is somewhat disappointing. I also believe this fact is not openly discussed in the text.

Some more comments regarding writing:
- From the title, it is not clear that the approach is applied to the music or image domain. Neither title nor abstract clarify that the proposed approach relies on time-series data.
- The abstract, intro and conclusion suggest that using (a) using physical symmetry for learning and (b) representation augmentation are two different contributions of the paper, when in fact, they refer to the same thing (i.e., the proposed idea).
- The term ''prior model'' is used in the abstract, but is not properly explained. In the paragraph starting on line 37, this finally gets explained, but the exposition is again quite convoluted. This would be much easier to understand if the applications used for the experiments (pitch sequences/moving ball) were introduced right at the start.
- The same goes for ''global invariant style code'', the meaning of which is completely unclear until one reads the appendix. Again, the discussion should be much more concrete and clearly mention what style and content mean in the different application scenarios.
- Section 2 (Intuition) is actively unhelpful. Instead of a clean explanation (using example applications!), we are given pompous and unscientific prose such as ''we regard symmetry in physical law as a general inductive bias of the human mind'' (line 74).
- Line 87 introduces the term ''linear pitch factor'', without clarifying the difference between that and simply the term ''pitch''. I suppose the term pitch is not used since the learned latent variables do not encode pitch on a natural (e.g. MIDI) scale, but this only becomes clear to the reader much later in the manuscript.
- In lines 110f, the reader suddenly learns that the model is a variational autoencoder! This should have been made plain in Section 3.1, or even earlier.

Overall, this is a borderline paper for me, with a tendency towards reject.

**Questions:**

- How important is L_rec, relative to L_prior? That is, can we turn off L_rec and still get similar results?
- In Equation 2 (and other equations), how do you use the binary cross entropy loss to compare spectrograms (or pictures)? This is very unusual, since spectrograms do not have a value range [0,1]. Is this applied in an element-wise fashion?
- Line 122: ''In other words, we add or subtract the latent codes by a random scalar.'' -- Unclear. Do you mean you add a scalar *to* the latent code?
- Line 137f: Is the choice of sequence (a scale upward and downward) relevant? Could you also choose a stationary sequence? Or a random sequence (transposed to different keys)?
- Lines 286ff: I think the limited usefulness of the VAE compared to the AE stems from the fact that the datasets used are so simple. There are simply no sources of variability that would need to be modeled by the Gaussian prior.

**Limitations:**

The limitations of the synthetic experimental setups should be discussed more openly in the text.

---

> ### Author Rebuttal · Authors · 2023-08-10
>
> Thanks for the detailed feedback and for raising those important questions. In the below chain of comments, we respond in a breakdown format.
>
> > **How important is L_rec, relative to L_prior? That is, can we turn off L_rec and still get similar results?**
>
> **Reply 1**:
> No, we can’t. L_rec is critical in preventing representation collapse.
>
> Your intuition to ablate L_rec is astute in a sense that L_prior is the conceptually central training objective — it literally evaluates the *latent dynamics*. However, without L_rec, the latent representation will collapse, where all possible x is encoded to the same z. In the case of collapse, the prior model will trivially learn the identity function to achieve L_prior=0. Additionally, collapse is guaranteed almost everywhere in the random initialization space since L_prior is a contra**c**tive loss that pulls representations inwards. L_rec forces z to contain at least the information required to reconstruct x, preventing collapse.
>
> > **In Equation 2 (and other equations), how do you use the binary cross entropy loss to compare spectrograms (or pictures)? This is very unusual, since spectrograms do not have a value range [0,1]. Is this applied in an element-wise fashion?**
>
> **Reply 2**:
> We apply it in a pixel-wise fashion. A binary cross entropy (BCE) loss can be used because we normalise the energy value of spectrograms to the range [0, 1] in advance (paper line 144). In fact, we have done experiments comparing using BCE vs. using L_2 loss, and yielded no difference in performance.
>
> > **Line 122: ''In other words, we add or subtract the latent codes by a random scalar.'' -- Unclear. Do you mean you add a scalar *to* the latent code?**
>
> **Reply 3**:
> Yes, we mean “we add a random scalar to the latent code”. We will update the paper accordingly.
>
> > **Line 137f: Is the choice of sequence (a scale upward and downward) relevant? Could you also choose a stationary sequence? Or a random sequence (transposed to different keys)?**
>
> **Reply 4**:
> We should be able to use random sequences transposed to different keys to train. In contrast, stationary sequences (if you mean all notes in a sequence share the same pitch) will not work Since the RNN would collapse.
>
> > **The results for the proposed approach show a slight or moderate improvement compared to the K=0 baseline. …(see, e.g., Figure 3, K=0 vs K=4).**
>
> **Reply 5**:
> It may be a bit misleading if we only focus on Fig 3 in order to understand the effectiveness of SPS. If we take a comprehensive view of the experimental results, i.e. considering Fig 3 alongside Fig 11, it becomes evident that the introduction of timbre factors makes the improvement (of the linearity of pitch) from physical symmetry much more pronounced. As the task difficulty increases, the role of symmetry becomes increasingly significant.
>
> > **From the title, it is not clear that the approach is applied to the music or image domain. Neither title nor abstract clarify that the proposed approach relies on time-series data.**
>
> **Reply 6**:
> We mention physical symmetry in the title and video and audio in the abstract, which implies what we deal with is time-series data. For the sake of clarity for readers, we will indicate time-series  in the revised abstract.
>
> > **The abstract, intro and conclusion suggest that using (a) using physical symmetry for learning and (b) representation augmentation are two different contributions of the paper, when in fact, they refer to the same thing (i.e., the proposed idea).**
>
> **Reply 7**:
> We do not think they are the same things though they are strongly related in this work. Using physical symmetry is a novel methodology to unsupervisedly learn interpretable representation from time-series data, while representation augmentation is just one way to implement it. Of course there might be other pure mathematical ways (as opposed to using imaginary samples) to enforce physical symmetry constraints but we have not found them yet. In addition, representation augmentation can also help methods other than physical symmetry that regularise the latent space. Therefore, we regard them as two different contributions.
>
> > **I think the limited usefulness of the VAE compared to the AE stems from the fact that the datasets used are so simple. There are simply no sources of variability that would need to be modeled by the Gaussian prior.**
>
> **Reply 8**:
> In general, VAE yields better representations than AE even when the data is simple and the Gaussian prior works very well even on non-Gaussian data. Our work indicates that physical symmetry could be an inductive bias which could 1) substitute the gaussian prior for some time-series tasks and 2) at the same time be more interpretable.
>
> > **Section 2 (Intuition) is actively unhelpful. Instead of a clean explanation (using example applications!).**
>
> **Reply 9**:
> The main contribution of the paper is to draw inspiration from modern physics to address a key challenge in machine learning: how to learn interpretable low-dimensional factors from unlabeled data without relying on domain-specific knowledge. Therefore, the writing in the beginning focuses a lot on high-level abstraction and our vision & intuition, which aims to help readers to analogously understand the big picture and why physical symmetry is a general inductive bias.
>
> Above all, an outstanding paper is not merely a tech report, but also a story with high-level and inspiring ideas. We hope the significance of this text could be reconsidered.

---

> > ### Comment · Reviewer_TYBC · 2023-08-14
> >
> > Thank you for your detailed responses to all reviews and for answering some of my questions!
> >
> > Overall, I think this paper has a very interesting idea at its core and I would not oppose accepting it.
> >
> > Some additional comments:
> > - Reply 1: Thanks for clarifying it. I suppose for the simple synthetic datasets used, most of the heavy lifting seems to be done by the reconstruction loss. It will be interesting to see in future work how much physical symmetry can contribute for more complicated scenarios and real-world datasets.
> > - Reply 7: OK. I would suggest further emphasizing the distinction between physical symmetry as "concept" and representation augmentation as "implementation".
> > - Reply 9: I suppose we fundamentally disagree on this. I believe claims in scientific texts should not go beyond what is shown in the results provided or supported by literature. The results in this paper certainly do not support any speculation about "the human mind".
> >
> > I hope that, even if we disagree on the subject of Reply 9, my comments on the writing might help to make the introduction a little bit clearer. In particular, consider emphasizing the application examples (pitch sequences/moving ball) and the autoencoder structure (i.e. reconstruction!) of the model.

---

> > > ### Author Response · Authors · 2023-08-15
> > >
> > > We would like to thank you again for the appreciation of our idea and the willingness to lean toward accepting it.
> > >
> > > Also, both the original critiques and additional comments helped us better understand the writing from a reader's perspective and we will certainly follow some important suggestions to make it clearer. In particular, trying to make it clear which part is pure conceptual comprehension and which part is implementation. In the revision, we will try to use a more rigorous expression to assist readers in comprehending physical symmetry through analogical reasoning.

---

### Official Review · Reviewer_mck6 · 2023-06-29

**Soundness:** 2 fair
**Presentation:** 3 good
**Contribution:** 2 fair
**Rating:** 5
**Confidence:** 4

**Summary:**

The paper presents a new way of training a self-supervised system that includes a data augmentation module in the latent space that leverages physical symmetry. This introduction of a transformation covariance in the latent space while training helps learn interpretable, robust and data-efficient representation. An evaluation is performed on synthetic data in two domains: music and videos. The evaluation shows that the new training system is able to learn a presentation that fits human perception.

**Strengths:**

1) The paper is very well written and very interesting
2) The idea of leveraging symmetry by enforcing latent space covariance seems quite novel to me and could have an impact in the field of Self-Supervised learning.
3) The method seems to perform as claimed on the presented evaluation (on toy datasets).

**Weaknesses:**

1) The evaluation is done only on toy problems with synthetic datasets.
2) Part of the evaluation is not very convincing for several reasons:
   a) First experiment:
- The task is too easy for the reported metrics: the R2 scores are saturated to 1, which makes the comparison with the baseline (SPICE) quite questionable.
- The ablation study (K=0) is a bit disappointing, as the improvement of using symmetry (K=4) doesn't seem significant. So the good properties of the learned representation seem more linked to the encoder/decoder architecture than to the use of symmetry covariance.
   b) sample efficiency: the correlation between K and sample efficiency is quite unclear in figure 6: for 512 and 2048 samples, K=4 and K=16 seem to have very similar performances.
3) It's not trivial how the dimension of the latent space and the symmetry should be chosen for problems a bit more complex than the toy ones presented in the evaluation.

**Questions:**

I really don't get why physical symmetry could be seen as a counterfactual inductive bias. Any latent representation model could be used to answer "what if" questions about the model (or rather the latent representation itself), which doesn't make them counterfactual inductive.
Could the author be more specific about what they mean here?

**Limitations:**

1) The link between the group of symmetry to be used for training and the actual task may not be straightforward for tasks a bit more elaborated than the ones proposed in the paper.

2) The claimed interpretability and the link with causality are questionable:

   a) Sample efficiency and extendability are very barely linked to interpretability

   b) The "interpretability" of the latent representation in the experimental part is largely due to design choices on the latent space and the symmetry, which can be considered as domain-knowledge priors. So I don't think the model helps generate interpretable representations but is only a way of encoding domain-knowledge priors. For instance, when the authors claim, "our model learns a more interpretable pitch than...", the model is actually not learning a more interpretable representation, but rather the prior was correctly handled by the symmetry. Same when they claim "... leads to a better enforcement of interpretability". It would be probably safer to talk about "better enforcement of the prior".
The example with the video has similar issues: implicitly, the chosen symmetry considers that the complicated dimension (height), which as a non-linear behavior, should be kept free, while the two others can be easily contained. This really sounds like a domain knowledge prior.

---

> ### Author Rebuttal · Authors · 2023-08-10
>
> We thank the reviewer for providing the feedback and raising important questions and concerns. We answer these questions here and also like to take the opportunity to address some concerns raised in weaknesses/limitations.
>
> > **Why could physical symmetry be seen as a counterfactual inductive bias?**
>
> **Reply 1**:
> We consider the physical symmetry as a counterfactual inductive bias on latent dynamics. In other words, it addresses the question if z_[1:t] is transformed, what z_[t+1:k] would be. Other approaches, like Variational Autoencoders (VAEs), do not consider the temporal aspect of the latent z. Therefore, we do not consider them to be counterfactual inductive biases. Thank you for bringing up this issue, which made us aware of the lack of clarity in our description. We will make sure to clarify this in the revised version.
>
> > **The choices on the latent space and the symmetry group assumptions sound like domain knowledge?**
>
> **Reply 2**
> We distinguish the difference between symmetry assumptions and domain knowledge through the following three aspects.
>
> **1) Representation augmentation requires much less knowledge than data augmentation.**
> Consider pitch extraction. Almost all related works use pitch-shift augmentation, but on the data and not on the representation. It requires the knowledge of how to transform the input data to shift the pitch. Simple techniques include time stretching and spectrogram frequency stretching, and advanced techniques (vocoders etc.) involve shifting harmonics without changing the timbre. In comparison, SPS just translates the representation vector.
>
> As for the vision task, it is not clear at all how to apply 3D spatial translation-rotation augmentation to unlabelled video. With our SPS method, one only needs to translate and rotate a representation vector of length 3.
>
> **2) Domain knowledge and representation symmetry belongs to two different conceptual levels.**
> When we pick group assumptions for a given problem, the kind of knowledge required is more abstract (and general) than what we usually call “domain knowledge”. It is us human’s beliefs about concepts themselves, not how concepts are encoded from data.
>
> Concretely, recall that music and vision are two vastly different fields. Their respective domain knowledge systems look nothing like each other. In comparison, SPS solves one task with 1D translation and the other with 2D translation-rotation. Their forms look overarchingly similar! To generalise, drastically different systems can share formally similar physical symmetries. The “expertise” involved in picking symmetries is utterly different from what “domain expertise” refers to.
>
> **3) We may not even need a correct symmetry assumption for successful representation learning with SPS.**
> The intuition section of our paper mentions that many modern physicists *start* with symmetry (in law) assumptions *and then* obtain testable theories of physical laws. Notice: first there are guesses about symmetry, and after that, “domain knowledge” is created.
>
> What it means is that symmetry is so fundamentally general that proposing symmetry assumptions is an efficient way of regularising concepts, even before we settle down with any domain knowledge. One is free to try multiple symmetry assumptions and see which ones learn better representations.
>
> Our experiments (section 5.3) already show that even with *incorrect symmetry assumptions* SPS can still learn to extract 3D Cartesian coordinates from the unlabelled bouncing ball dataset.
>
> > **Part of the evaluation is not very convincing，R2 scores are saturated to 1, which makes the comparison with the baseline (SPICE) quite questionable. The ablation study (K=0) is a bit disappointing, as the improvement of using symmetry (K=4) doesn't seem significant. So the good properties of the learned representation seem more linked to the encoder/decoder architecture than to the use of symmetry covariance.**
>
> **Reply 3**:
> The $R^2$ score may be a bit misleading if we only focus on Table 1. If we take a comprehensive view of the experimental results, i.e. considering Table 1 alongside Figure 12, it becomes evident that the introduction of timbre factors makes the $R^2$ improvement from physical symmetry even more pronounced. As the task difficulty increases, the role of symmetry becomes increasingly significant.
>
> > **sample efficiency: the correlation between K and sample efficiency is quite unclear in figure 6: for 512 and 2048 samples, K=4 and K=16 seem to have very similar performances.**
>
> **Reply 4**:
> We are not trying to show a linear relation between K and interpretability (linear project loss). The point is that using representation augmentation is better than not using it, regardless of the dataset size. Additionally, it can be observed that increasing the amount of data makes it easier for the model to learn interpretable representations, so the interpretability provided by representation augmentation saturates when the data size reaches 2048. Furthermore, we are unable to determine the optimal value of K for different dataset sizes. We will consider revising the original text to clearly highlight this point, thank you for bringing it up.

---

> > ### Comment · Reviewer_mck6 · 2023-08-17
> > **Answer to rebuttal**
> >
> > Dear authors,
> > Thank you for carefully reading my review and answering my comments. Reply 3 and 4 clarified my understanding of your experimental claims.
> >
> > I'm still puzzled about the physical symmetry being responsible for counterfactual inductive bias, while any method that would predict future samples of a time series from past ones could be considered counterfactual (while the underlying causal mechanism is usually quite questionable).
> > Why the use of physical symmetry would make the system "more causal"?
> >
> > I'm still unconvinced about the distinction between domain knowledge and physical symmetry design. I still think that to choose your symmetry, you still need implicit domain knowledge. For both experiments, you do use some sort of domain knowledge about the underlying data to pick the symmetries: for instance, for the videos, the two horizontal coordinate dimensions have a different role than the height, domain knowledge that you somewhat use to pick a different symmetry on the two first latent dimensions and on the third one. And this seems necessary to keep a bit of interpretability.
> > Though it may only be a question of your definition of "domain knowledge," which seems to be a bit more restrictive than mine.
> >
> > Overall, I still think the paper is an interesting contribution that is worth to be published.

---

> > > ### Author Response · Authors · 2023-08-21
> > >
> > > Thank you for your comments. We hope the following explanations would make what puzzled you clearer.
> > > According to Pearl and Mackenzie [1], the “Ladder” of Causality has three levels:  Seeing (statistics), Doing (intervention), and Imagining (counterfactuals). A higher level means “more causal”. If the model only learns statistical correlation between the past and the future, it is at level 1 – Seeing, which is pure statistics. Many time-series models (e.g., vanilla HMM and RNN) are at this level. Intervention is about doing — observing what happens when changes are made. Counterfactuals can be thought of as "imaginings". It allows us to consider specific scenarios that are inexpressible at level 1 or 2. E.g., for the video experiment, training with physical symmetry means the model can imagine how changes of the past would affect changes in the future (I.e. $(R(S(z_{1:T}))) = S(z_{T+1})$) for each trajectory. By training with representation augmentation, our learned model can predict and reconstruct unseen trajectories, through the translation or rotation transformation on its latent $z$. Therefore, physical symmetry transcends our time-series model from level 1 to level 3, and we consider such inductive bias counterfactual.
> > >
> > > Regarding domain knowledge, we agree that there may be no strict lines to distinguish between domain vs. non-domain knowledge. Our point is that physical symmetry yields much looser assumptions on the underlying data. Let’s take the video case as an example, the only strict assumption here is that the underlying representation of each data sample is low-dim — 3D. Here, we need ​​a comprehensive view of the experimental results. From Figure 8, we see that even with incorrect symmetry assumptions, the model can still learn interpretable 3D cartesian space. Hence, the “correct” symmetry assumption (which seems to incorporate the domain knowledge of a real 3D cartesian world) is actually not necessary. Additionally, the 3D assumption actually doesn’t even imply the domain knowledge of a 3D *cartesian* coordinate. In our early experiments (which we didn’t put in the paper), we accidentally made each trajectory start from the same location. As a result, the model learned a cylindrical coordinate system (in which the first 2d is a polar coordinate).
> > >
> > > Therefore, physical symmetry is really an inductive bias tailored for the latent space, which puts minimum assumption on the data (of a specific domain). If there is anything implicit, it’s fair to say that physical symmetry implies both visual domain and auditory domain in the same fashion, so it is a more generalized inductive bias.
> > >
> > > [1] Pearl, J., & Mackenzie, D. (2020). The book of why: the new science of cause and effect. Basic books.

---

### Official Review · Reviewer_dLV3 · 2023-07-06

**Soundness:** 3 good
**Presentation:** 4 excellent
**Contribution:** 3 good
**Rating:** 7
**Confidence:** 4

**Summary:**

This paper utilizes the concept of physical symmetry as a self-supervised constraint within an auto-encoder framework to enhance the learning of interpretable and disentangled representations. The authors validate their approach through experiments conducted on unlabelled monophonic music audio and monocular videos featuring a bouncing ball. The results demonstrate high projection accuracy and successful disentanglement of style and content. Additionally, the paper introduces representation augmentation techniques that improve sample efficiency and enhance interpretability.

**Strengths:**

1. This paper is well-written and easy to understand, demonstrating a cohesive and structured presentation of the research.
2. Convincing experiment results, which align with the stated objectives, provide strong support for the findings of this study.
3. The representation augmentation technique is novel and supported by empirical evidence showcasing its effectiveness.
4. The results on color-texture disentanglement is clear and interpretable

**Weaknesses:**

1. The experimental setup seems to be somewhat simplistic despite the experimental results being good
2. Only few experiments on real-world dataset are provided, which is not able to fully demonstrate the generalizability and applicability of your method.
3. There is a lack of theoretical evidence of your proposed method and further explanation of the relationship between physical symmetry and representation learning.

**Questions:**

1. Could you explain what a random-pooling layer does and the details of how you split the style and content vector? Have you ever tried other methods to disentangle the style and content part?
2. Will the sequence length (the value of T) affect the final result?
3. Could you describe any unique aspects or challenges of the dataset you have proposed?

**Limitations:**

1. The generalizability and applicability for real-world dataset may be limited.
2. The proposed dataset should be more challenging for further utilization on research

---

> ### Author Rebuttal · Authors · 2023-08-10
>
> We thank the reviewer for the positive review! We really appreciate the concise summarisation on both the strengths and the current limitation of the paper. We hereby respond in a breakdown format.
>
> > **Could you explain what a random-pooling layer does and the details of how you split the style and content vector? Have you ever tried other methods to disentangle the style and content part?**
>
> **Reply 1**:
> Random pooling simply means we randomly pick a vector $z_{i, s}$ as $z_{\tau, s}$ from the style factor sequence $\textbf{z}_{1:T, s}$.  This operation encourages the learned style representation to be constant over time. Also, To split the style and content vector we first cut the vector $z$ into two parts. If $z$ has $n$ dimensions, we choose $z[0:m]$ as $z_c$ and $z[m:n]$ as $z_s$, where $n$ and $m$ are hyperparameters (For the audio problem $m=1$ and $n=2$. For the video problem $m=3$ and $n=2$).
>
> As for other methods of disentanglement, we haven’t tried any since a random pooling (our first try) simply works within an SPS+ framework. We certainly note that the content-style definition imposed by SPS+ is very simple — content refers to representations that change over time and adhere to physical symmetries, while style refers to representations within the same sequence that remain unchanged over time (global invariance). In general, there can be meaningful factors that change over time but whose patterns of change do not adhere to physical symmetries. In such cases, we propose to combine SPS with other methods for decoupling in the future. For instance, applying SPS to Causal Variational Autoencoders (CVAEs), disentangled GANs, or contrastive/non-contrastive methods. Actually, an ongoing experiment of our team is to combine VICReg [1] and SPS to obtain more interpretable content factors, and preliminary results suggest that such combination outperforms either SPS or VICReg alone on some non-trivial cases.
>
> [1] Bardes, A., Ponce, J., & LeCun, Y. (2021). Vicreg: Variance-invariance-covariance regularization for self-supervised learning. arXiv preprint arXiv:2105.04906.
>
> > **Will the sequence length (the value of T) affect the final result?**
>
> **Reply 2**:
> The sequence length T has no direct effect on learning interpretable factors, but has an indirect effect. A sufficiently long sequence can ensure that the prior model (RNN) learns meaningful system dynamics, and this is the premise for physical symmetry constraints to work. If the prior model can already learn the correct system dynamics with a small T, increasing the sequence length will no longer help learn interpretable representations.
>
> > **Could you describe any unique aspects or challenges of the dataset you have proposed?**
>
> **Reply 3**:
> Due to the small size of our dataset, it is challenging for existing unsupervised learning methods to learn meaningful interpretable representations.
> The audio dataset (as described in A.4.1) contains *only* 2400 audio clips played by multiple instruments. In the computer music domain, we know how hard it is to unsupervisedly 1) disentangle pitch and timber, and 2) learn a linear pitch concept. As far as we know, all prior works incorporate strong domain knowledge, such as pitch shifting for pseudo-label generation [1, 2] or using instrument labels [3].
>
> Likewise, learning concepts such as 3D coordinates from unlabelled video has long been a far-fetched fantasy for CV researchers. To be fair, in recent years we have seen exciting progress on this front. For example, LEAP [4] performs physically meaningful representation disentanglement via causal discovery. But even LEAP *requires* not only independent noises but also a sufficient causal structure (e.g., five or more balls interacting in the same scene via springs forces) in order to learn disentangled location factors.
>
> [1] Luo, Y. J., Cheuk, K. W., Nakano, T., Goto, M., & Herremans, D. (2020, October). Unsupervised Disentanglement of Pitch and Timbre for Isolated Musical Instrument Sounds. In ISMIR (pp. 700-707).
> [2] Gfeller, B., Frank, C., Roblek, D., Sharifi, M., Tagliasacchi, M., & Velimirović, M. (2020). SPICE: Self-supervised pitch estimation. IEEE/ACM Transactions on Audio, Speech, and Language Processing, 28, 1118-1128.
> [3] Luo, Y. J., Agres, K., & Herremans, D. (2019). Learning disentangled representations of timbre and pitch for musical instrument sounds using gaussian mixture variational autoencoders. arXiv preprint arXiv:1906.08152.
> [4] Yao, W., Sun, Y., Ho, A., Sun, C., & Zhang, K. (2021). Learning Temporally Causal Latent Processes from General Temporal Data. arXiv preprint arXiv:2110.05428.

---

> > ### Comment · Reviewer_dLV3 · 2023-08-18
> >
> > I greatly appreciate your comprehensive review and the thoughtful responses provided to my questions. Your explanations are both clear and well-justified. The proposed method is straightforward and supposed to be great, and I'm still looking forward to the potential for even more interesting frameworks for manipulating the style and content vectors.
> >
> > In summary, your paper introduces a novel and effective method, presented within a well-organized structure and expressed with well-crafted sentences. I believe it is a good paper deserving of acceptance.

---

> > > ### Author Response · Authors · 2023-08-21
> > >
> > > We are glad our responses have contributed to your evaluation and confidence in our work. Thank you once again for your support.

---

### Official Review · Reviewer_5wLb · 2023-07-06

**Soundness:** 4 excellent
**Presentation:** 4 excellent
**Contribution:** 3 good
**Rating:** 7
**Confidence:** 4

**Summary:**

This paper proposes a symmetry equivariance constraints on the transition dynamics of latent representations for time-indexed data. The claim is that imposing these certain symmetries create interpretable model representations that correspond to popular domain-specific representations in the audio and video domains.

Experiments on audio demonstrate how enforcing a translational symmetry recovers an interpretable one-dimensional latent factor corresponding to pitch. Experiments on video show how enforcing translational and rotational symmetries recovers interpretable three-dimensional latent factors corresponding to spatial coordinates. Ablations also show some robustness of learning to mis-specification of symmetries.

**Strengths:**

This is an inspiring paper! The paper is well-motivated and well-written. The methodology is clearly described and easy to follow (although I do think there might be an even better probabilistic formulation/presentation of the training objective; see the Questions section). The methods make sense and I feel confident that I could implement these ideas myself, based on the description in the paper.

The experiments are well-executed and support the hypothesis that physical symmetries can provide a powerful inductive bias, at a higher level of abstraction than more domain-specific approaches.

**Weaknesses:**

My only major criticism is that the experiments are in somewhat "toy" settings. It would be interesting to see an application of these ideas to a more significant problem, with stronger baselines.

A very minor criticism of the (otherwise excellent!) introduction: "Such an approach is very different from human learning; even without formal music training, one can at least perceive pitch, a fundamental music concept, from the experience of listening to music." I think this claim is too strong. It is not clear how much of human perception is learned from experience, vs. baked in to our genetics and brain structure. This claim isn't important to your central argument, so I suggest moderating it a bit.

**Questions:**

Is it possible to frame the training objective (Equation 1) as a proper probabilistic loss? See [1] (the static case) and [2] (the time-indexed case) for the probabilistic formulations of the Gaussian prior. If this is possible, it might help to clarify why/whether it is possible to drop the KL regularization term.

Does this work relate to previous work on imposing symmetries on intermediate layers of neural networks? E.g. [3] and [4]. I am not deeply familiar with that line of work, but I'm bringing it up because it might be relevant (and at the very least, you might find it interesting if you aren't already aware).

[1] Auto-Encoding Variational Bayes. Diederik P. Kingma, Max Welling.

[2] A Recurrent Latent Variable Model for Sequential Data. Junyoung Chung, Kyle Kastner, Laurent Dinh, Kratarth Goel, Aaron Courville, Yoshua Bengio.

[3] Deep Symmetry Networks. Robert Gens, Pedro Domingos.

[4] Group Equivariant Convolutional Networks. Taco S. Cohen, Max Welling.

**Limitations:**

The experiments presented in this work use low-dimensional latent spaces: 1 dimension for audio and 3 dimensions for video. It is not clear how to adapt these methods to the high-dimensional latent spaces commonly used (and often required) for expressive models where model performance and capacity is prioritized more highly than interpretability. Furthermore, it is not clear what symmetries we ought to impose upon high-dimensional latent spaces. That said, these questions are clearly beyond the scope of the present work: I see this limitation more as an interesting avenues for future investigation rather than a weakness of the present work.

---

> ### Author Rebuttal · Authors · 2023-08-10
>
> Thank you for giving the insightful feedback and raising such important questions. Below we respond in a breakdown format.
>
> > **Is it possible to frame the training objective (Equation 1) as a proper probabilistic loss? See [1] (the static case) and [2] (the time-indexed case) for the probabilistic formulations of the Gaussian prior. If this is possible, it might help to clarify why/whether it is possible to drop the KL regularization term.
> [1] Auto-Encoding Variational Bayes. Diederik P. Kingma, Max Welling.
> [2] A Recurrent Latent Variable Model for Sequential Data. Junyoung Chung, Kyle Kastner, Laurent Dinh, Kratarth Goel, Aaron Courville, Yoshua Bengio.**
>
> **Reply 1**:
> Thank you for referencing [1] [2] to help us in obtaining a sound probabilistic formulation for SPS. We have been working on the theoretical grounding for SPS, but to be very frank there are no concrete results yet. We’d like to share our current direction: First, assume the dataset follows a certain symmetry. Then, show that including the representation augmentation terms in the training objective is a way of imposing a prior and maximising the likelihood of the dataset. Specifically, given the symmetry assumptions, rewrite the likelihood of the dataset to include a lower bound that is maximised by representation augmentation training (following the assumed symmetry). We will greatly appreciate your feedback in this regard! We look forward to the discussion phase and consider it as an opportunity to collaboratively discover a probabilistic formulation, in which case we will eventually and duly acknowledge your input.
>
> > **Does this work relate to previous work on imposing symmetries on intermediate layers of neural networks? E.g. [3] and [4]. I am not deeply familiar with that line of work, but I'm bringing it up because it might be relevant (and at the very least, you might find it interesting if you aren't already aware).
> [3] Deep Symmetry Networks. Robert Gens, Pedro Domingos.
> [4] Group Equivariant Convolutional Networks. Taco S. Cohen, Max Welling.**
>
> **Reply 2**:
> Thank you for referencing [3] [4] as they are very exciting to read and indeed very related to learning with symmetry. [3] [4] are in the same line with [5] [6] (referenced in the paper); they emphasise some equivariant relations **between signal x and representation z**: When a certain transformation is applied to x, z should keep invariant or follow a similar/same transformation. Specifically, [3] [4] develop network architectures invariant to continuous group transformations of input data and intermediate feature maps. On the contrary, SPS **deals with z only**, and its time-series equivariance. Our paper’s major contribution is to “escape” the signal x space and only prescribe constraints in the representation z space.
>
> There’s actually another reason we appreciate your references so much: [3]’s review on Lie groups and object orbits (section 2) gave us new insights to our method. [3] is concerned with classification tasks, characterised by their VC dimension and sample complexity, which they show can be reduced by using symmetry if object orbits are homogeneously labelled. That is analogous to our method if we consider energy-based models (EBMs) of time series. An EBM outputs an energy (scaler) given one trajectory z[1:T] which, intuitively, denotes how “physically realistic” the EBM assesses the trajectory to be. This way, the energy forms object orbits under specific symmetry assumptions, and we require each object orbit to have homogenous energy. This analogy may eventually enable new theoretical groundings for our formulation of SPS. As such we are extra thankful that you referenced [3] [4].
>
> [5] Dupont, E., Martin, M. B., Colburn, A., Sankar, A., Susskind, J., & Shan, Q. (2020, November). Equivariant neural rendering. In International Conference on Machine Learning (pp. 2761-2770). PMLR.
> [6] Sanghi, A. (2020). Info3d: Representation learning on 3d objects using mutual information maximization and contrastive learning. In Computer Vision–ECCV 2020: 16th European Conference, Glasgow, UK, August 23–28, 2020, Proceedings, Part XXIX 16 (pp. 626-642). Springer International Publishing.
>
> > **A very minor criticism of the (otherwise excellent!) introduction: "Such an approach is very different from human learning; even without formal music training, one can at least perceive pitch, a fundamental music concept, from the experience of listening to music."I think this claim is too strong. It is not clear how much of human perception is learned from experience, vs. baked in to our genetics and brain structure. This claim isn't important to your central argument, so I suggest moderating it a bit.**
>
> **Reply 3**:
> Thank you for your kind advice. We recognize that such a strong claim calls for supportive evidence, but the psychological experiments that would prove/disprove our claim are not there yet to our knowledge. We will consider removing/soften the claim for the sake of soundness — thank you for bringing it up.

---

> > ### Comment · Reviewer_5wLb · 2023-08-13
> >
> > > We’d like to share our current direction: First, assume the dataset follows a certain symmetry. Then, show that including the representation augmentation terms in the training objective is a way of imposing a prior and maximising the likelihood of the dataset. Specifically, given the symmetry assumptions, rewrite the likelihood of the dataset to include a lower bound that is maximised by representation augmentation training (following the assumed symmetry).
> >
> > Yes, this is approximately what I had in mind. My point was that the KL divergence term of the VAE loss arises naturally by from defining a proper probability distribution over the join distribution p(x,z) on observations x and latent variables z, and constructing a lower bound to tractably optimize the marginal distribution p(x). In the simplest setting, the prior p(z) is chosen to be Gaussian. It seems possible to me (I admittedly have not fully thought through the idea) that, by encoding a symmetry structure into your prior over the latent sequence, the lower-bound derived by the usual VAE argument would correspond to your Equation 1.
> >
> > To be very clear: the above discussion is beyond the scope of the reviewing process. **I think this is a good paper and that it should be accepted** in its current form (+ any revisions to address other reviewers' concerns). Formalizing SPS in a probabilistic framework would be an interesting subject for future work.

---

> > > ### Author Response · Authors · 2023-08-15
> > >
> > > > It seems possible to me (I admittedly have not fully thought through the idea) that, by encoding a symmetry structure into your prior over the latent sequence, the lower-bound derived by the usual VAE argument would correspond to your Equation 1.
> > >
> > > Thank you for your insight on formulating SPS in a probabilistic framework! We find it a good place to start formalizing exisitng intuitions. We agree that it is a subject for future work.
> > >
> > > Thank you for your detailed study of our work and the constructive, positive review!

---

### Official Review · Reviewer_wTJe · 2023-07-14

**Soundness:** 4 excellent
**Presentation:** 4 excellent
**Contribution:** 4 excellent
**Rating:** 7
**Confidence:** 5

**Summary:**

This study introduces a new methodology to learn interpretable representations from data by incorporating physical symmetry as a self-consistency constraint in the latent space. It addresses a key challenge in machine learning: how to learn interpretable low-dimensional factors from unlabeled data without relying on domain-specific knowledge. The method, named Self-Supervised learning with Physical Symmetry (SPS), uses the concept of physical symmetry to ensure that the model's learned dynamics are invariant under certain transformations.

The research applies SPS in two domains: music and computer vision. In music, SPS successfully learns a linear pitch factor from unlabeled monophonic audio, and in computer vision, it learns a 3D Cartesian space from unlabeled videos of a simple moving object. The study suggests that the use of physical symmetry could lead to a new technique called representation augmentation, which enhances the model's sample efficiency.

The authors draw inspiration from the approach in modern physics where scientists often start from a symmetry assumption to derive laws and predict properties of fundamental particles. This idea of symmetry as a fundamental guiding principle is being used as an "inductive bias" for their representation learning model, helping to create an interpretable low-dimensional latent space.

They view symmetry in physical law not only as a design principle of nature, but also as a cognitive bias of the human mind. This suggests that they believe the learning model should reflect the same intuitive understanding of symmetry.

Physical symmetry is also used as the basis for a learning technique called "representation augmentation". This involves generating additional pairs of training samples from existing ones by applying certain group transformations, which are informed by the symmetries. This process helps to improve sample efficiency and imposes a regularization effect on the learning model.

Finally, through the lens of causality, the authors describe physical symmetry as a counterfactual inductive bias. This means the learning model is designed to ask "what if" questions, specifically what would happen if predictions were based on transformed latent codes. This concept also provides constraints on the model's encoder and decoder components, since they are trained end-to-end, keeping the overall structure and function of the model consistent with the underlying symmetry principle.

Key Objectives:

To introduce and explore the use of physical symmetry as a self-consistency constraint in the latent space of time-series data.

To develop the SPS methodology that applies physical symmetry to the prior model in an encoder-decoder framework.

To demonstrate the application of SPS in learning a linear pitch factor from unlabeled monophonic music audio, without any domain-specific knowledge about pitch scales.

To apply the SPS methodology in the computer vision domain, specifically learning a 3D Cartesian space from unlabeled videos of a simple moving object.

To examine the desirable properties of SPS, including conciseness, sample efficiency, robustness, and extendability.

To prove that even with an incorrect symmetry assumption, SPS can still learn more interpretable representations than baseline models.

To demonstrate the possibility of combining SPS with other learning techniques, allowing for content-style disentanglement from temporal signals.

**Strengths:**

Inductive Bias - By leveraging physical symmetry as an inductive bias, the learning model aims to uncover meaningful and interpretable low-dimensional representations from high-dimensional data. This approach aids the model in abstracting critical features of the data, such as the pitch of music notes or the 3D location of a moving object, while filtering out less relevant information.

Regularization and Model Robustness - The method uses symmetry-based representation augmentation and transformations, which generate additional training samples. This augmentation process improves sample efficiency and imposes regularization on the model, reducing overfitting. It also helps in maintaining consistency in predictions, thereby improving model robustness.

The model is designed with self-consistency in mind. The prior model's predictions for a transformed version of the latent space and the original version should be close, and so should their decoded reconstructions. This self-consistency constraint enhances the regularization of the latent space.

Training Objectives - The total loss function used in the model is a combination of several loss terms including reconstruction loss, prior prediction loss, symmetry-based loss, and KL divergence loss. This diversified loss function enables the model to optimize different aspects of the learning process, contributing to a better overall model performance.

Flexibility and Generality - This approach is versatile and can be applied to different problems. Different group transformations are used for different problems, demonstrating the method's ability to adapt to different contexts.

Variants - The model has different versions like SPSVAE and SPSAE, providing flexibility based on the specific needs of the task at hand. For instance, SPSAE could be used when a simpler model architecture without the KL divergence term is sufficient.

**Weaknesses:**

Dependence on Correct Symmetry Assumptions: While the experiments show that the method is robust to incorrect symmetry assumptions, choosing the right symmetry transformation S is crucial for optimal results. Incorrect assumptions can still lead to an inferior performance. Real-world data may contain complex symmetries not known a priori, which could lead to difficulty in correctly specifying these assumptions.

Dependence on Augmentation Factor: The efficiency and effectiveness of the model are significantly affected by the augmentation factor K. Choosing an optimal K is a challenge and requires extensive experiments. A poorly chosen K could result in less efficient training or lower performance.

High-Dimensional Latent Space: While the method has been demonstrated on 1D and 3D latent spaces, its effectiveness in higher-dimensional latent spaces is not fully explored. Real-world data can often have high-dimensional latent spaces, and it may be challenging to maintain the same level of performance in these cases.

Overfitting: While representation augmentation can increase the robustness of the model, applying it excessively might lead to overfitting, especially when data is scarce.

**Questions:**

How do the authors propose this approach would be scalable for other tasks and what are the ways to prevent overfitting with the approach?

SPS can still learn interpretable representations even with incorrect symmetry assumptions. What might be the theoretical basis for this? Could there be instances where this doesn't hold true?

How does the proposed SPS model relate to, or differ from, other models or techniques used in representation learning? Could SPS be integrated with other methods to address its limitations?

**Limitations:**

Authors do inform about the range of limitations for the work

Partial Information Capture: The authors note that when the underlying concept following physical symmetry only encompasses a part of the information present in the time series data and cannot fully reconstruct the inputs, SPS may not work effectively. This issue is potentially related to content-style disentanglement.

Inability to Distill Concepts from Multibody Systems: The current model struggles with learning concepts from systems with multiple interacting components. As examples, the authors mention that SPS fails to learn the concept of pitch from polyphonic music, or understand 3D space from videos featuring multiple moving objects.

Lack of Formal Theory for Quantification: The authors highlight the need for a formalized theory to quantify the impact of representation augmentation. Such a theory could help measure the degree of freedom in the latent space both with and without physical symmetry, and help explain why even incorrect symmetry assumptions can still lead to correct and interpretable latent space.

---

> ### Author Rebuttal · Authors · 2023-08-10
>
> Thank you for the review. We sincerely appreciate your comprehensive understanding of this paper’s vision and strengths as well as its current limitations. Additionally, thank you for the in-depth questions. We’d like to address them here to the best of our knowledge.
>
> > **How do the authors propose this approach would be scalable for other tasks**
>
> **Reply 1**:
> If a dynamic system has a low-dim intrinsic latent space (similar to the 1D pitch or 3D location tasks), we expect SPS to generalise well. For tasks with *high-dim* latent factors, as the review has pointed out, applying SPS may be challenging. Similar to modern physics, trying multiple symmetry hypotheses becomes the first step. For that, we foresee an “auto-SPS” in the future which automatically *searches* for symmetry hypotheses given the search scope. Since figure 8 shows that a range of symmetry assumptions may all lead to interpretable representations, such a search has a higher probability to hit relevant symmetry assumptions.
>
> As for tasks where physical symmetry alone is not strong enough to regularise concepts, we will need to combine SPS with other methods. We already have a simple example in our appendix: SPS+, which takes advantage of the temporally invariant style factor to enable SPS’s effect on the content factors. Alternatively, if the style factors are intractable and we do not aim to model the style factors’ temporal dynamics, we can use SPS with VICReg [1] to obtain more interpretable content factors, and preliminary results suggest that such combination outperforms either SPS or VICReg alone on some non-trivial cases.
>
> >**What are the ways to prevent overfitting with the approach?**
>
> **Reply 2**:
> Thank you for raising such a profound question. For SPS, overfitting may be caused jointly by insufficient data and the symmetry assumption. With a limited amount of data, there may be multiple ways of describing the system dynamics, and all of them may conform to the assumed symmetry, but some yield more natural representations than others. We encountered such a situation before. In the bouncing ball experiment, if all trajectories in the dataset overlap at a single point, SPS will treat that point as the origin and learn polar coordinates to represent the ground plane instead of Cartesian coordinates. Given the special dataset, polar coordinates fully adheres to the prescribed symmetry constraint. If the goal is to learn a unique interpretable representation without obtaining more data domains, the only way may be to add new symmetry constraints. However, interpretable representations are not unique to begin with. Even if the model doesn’t yield the expected representations, it has "discovered" new interpretable representations (e.g. polar coordinates), which can still be valuable. Analogously, if relativity represents the true system dynamics, then the Newtonian mechanics essentially overfits to macroscopic low-speed phenomena, yet it yields interpretable representations.
>
> > **SPS can still learn interpretable representations even with incorrect symmetry assumptions. What might be the theoretical basis for this? Could there be instances where this doesn't hold true?**
>
> **Reply 3**:
> That is an important question and we thank the reviewer for emphasising it. Our preliminary ideas include using measurement theory to characterise learnable representation dynamics with and without SPS. The correct symmetry would be the strictest equivariance constraint that is physically true. Other constraints are either unphysical or loose. Therefore, the theory that predicts the effect of incorrect symmetry assumptions can very possibly be the same theory that explains the efficacy of SPS itself, showing exactly how SPS constrains the representation dynamics. However, for a full theory please expect future work. It will be much appreciated if you would point us some directions during the discussion phase!
>
> While the rigorous mathematical theory behind the efficacy of SPS remains an open question, we can still consider SPS training with incorrect symmetry as a general regularisation problem in machine learning. A toy analogy is ridge regression – we know it leads to better results if the true prior is Gaussian. However, if we assume an incorrect mean, variance, penalty weight, or even to use lasso instead of L2 norm, it can still bring some benefits as long as the incorrect assumption is not “too far” from the true one. Again, to quantify “too far”, we will need a formal theory, and in Figure 8, we consider the cases to be not “too far”.
>
> > **How does the proposed SPS model relate to, or differ from, other models or techniques used in representation learning? Could SPS be integrated with other methods to address its limitations?**
>
> **Reply 4**:
> From a energy-based model point of view, SPS is a regularised, non-contrastive method. It is not an architectural method, making it compatible with other techniques. It is not a contrastive method since SPS doesn’t need contrastive samples. It is highly related to VICReg [1]; both try to find a general regularised inductive bias for the latent space, but VICReg maximises information content of predictable representations while SPS purely considers the dynamics of z time-series.
>
> Since representation augmentation is generally agonistic to the specific training pipeline, SPS can be integrated with other methods to address its limitations. In the paper we already used SPS in conjunction with VAE. In our later experiments, we try to solve an “unconstrained style factor problem” (related to A.3) by contrastive training with SPS+VICReg. VICReg is in charge of preventing representation collapse without any decoder and SPS regularises the content factor.
>
> [1] Bardes, A., Ponce, J., & LeCun, Y. (2021). Vicreg: Variance-invariance-covariance regularization for self-supervised learning. arXiv preprint arXiv:2105.04906.

---

### Decision · Program_Chairs · 2023-09-21

**Decision:**

Accept (poster)

**Comment:**

Dear Authors,

After extensive deliberation and a comprehensive review of the feedback provided by our esteemed reviewers, we have reached a decision concerning the submission. The paper brings forth a strong and captivating idea that warrants attention in the academic community. Nonetheless, it is evident that certain concerns have remained unaddressed, notably the paper's lack of clarity on the capability of the SPS to learn interpretable representations, especially when incorrect symmetry assumptions are present.

The scalability of the approach to diverse tasks has been a recurrent concern, and it would have been beneficial to delve deeper into mechanisms to circumvent overfitting. On a constructive note, the discussion surrounding the KL divergence term of the VAE loss, and the potential to encode a symmetry structure into the latent sequence, suggests exciting avenues for future work. The concerns raised about the intertwining of domain knowledge and physical symmetry design, especially in the context of the experiments, demand more nuanced articulation, perhaps by delineating the concept of 'physical symmetry' from its implementation. Lastly, while the use of physical symmetry as a causal inducer remains a point of contention, we believe that further exploration in this direction might provide clarity. The additional feedback on emphasizing the application examples and the autoencoder structure is well-taken.

Overall, given the potential and significance of the core idea, and with the expectation that the authors will address the highlighted concerns, we are inclined to accept the paper. We believe that the necessary revisions will elevate the paper's value and coherence.

Best reagards,

Area Chair